# DATASET DISTILLATION VIA COMMITTEE VOTING

## ABSTRACT

Dataset distillation aims to synthesize a smaller, representative dataset that preserves the essential properties of the original data, enabling efficient model training with reduced computational resources. Prior work has primarily focused on improving the alignment or matching process between original and synthetic data, or on enhancing the efficiency of distilling large datasets. In this work, we introduce **C**ommittee **V**oting for **D**ataset **D**istillation (CV-DD), a novel and orthogonal approach that leverages the collective wisdom of multiple models or experts to create high-quality distilled datasets. We start by showing how to establish a strong baseline that already achieves state-of-the-art accuracy through leveraging recent advancements and thoughtful adjustments in model design and optimization processes. By integrating distributions and predictions from a committee of models while generating accurate soft labels, our method captures a wider spectrum of data features, reduces model-specific biases and mitigates distributional shifts between synthetic data and original data. This voting-based strategy not only promotes diversity and robustness within the distilled dataset but also significantly reduces overfitting, resulting in improved performance on post-eval tasks. Extensive experiments across various datasets and IPCs (images per class) demonstrate that Committee Voting leads to more reliable distilled data compared to single/multi-model distillation methods, while also generalizing to non–training based frameworks and more challenging tasks such as synthetic-to-real transfer.

## 1 INTRODUCTION

The rapid growth of large datasets has significantly advanced computer vision and deep learning applications, enabling models to achieve high accuracy and generalization across diverse domains. However, training on massive datasets presents challenges such as high computational cost, memory usage, and long training times, especially for resource-constrained environments. To address these issues, *dataset distillation* has emerged as an effective technique to condense large datasets into smaller, representative sets, allowing for efficient model training with minimal performance loss. Despite its promise, a key challenge remains: *capturing the essential features of the original data while avoiding overfitting to specific patterns or noise.*

Prior dataset distillation methods (Wang et al., 2018; Yin et al., 2023; Sun et al., 2024; Yin & Shen, 2024) often rely on single-model frameworks that may struggle to generalize across complex, diverse datasets and architectures. These approaches can introduce biases specific to the model used, resulting in distilled datasets that may not fully capture the richness of the original data. To overcome these limitations, we propose **C**ommittee **V**oting for **D**ataset **D**istillation (**CV-DD**), a framework that lever-

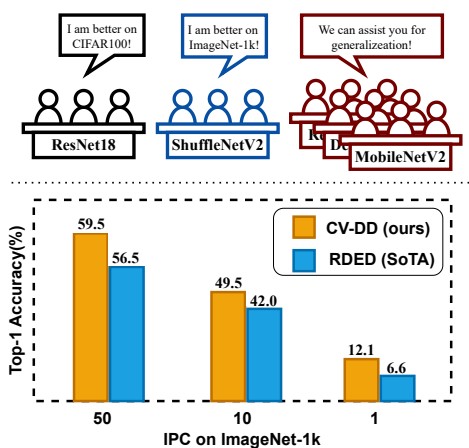

Figure 1: Top illustrates the motivation of our committee voting-based dataset distillation, highlighting its ability to reduce bias from individual model knowledge. Bottom shows the performance improvement over previous state-of-the-art method RDED (Sun et al., 2024) on ResNet-18.

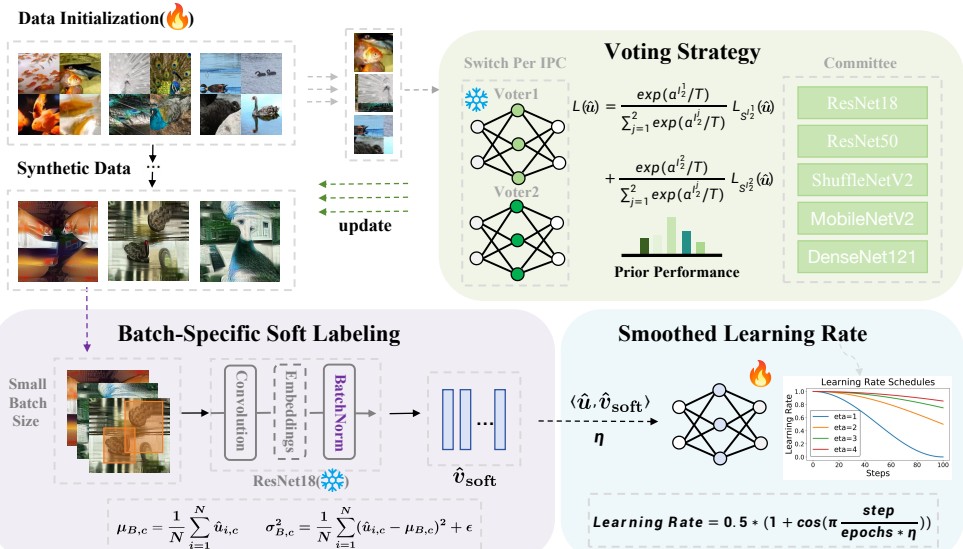

Figure 2: The process begins with **Data Initialization** to generate synthetic data from the original data distribution. In **Voting Strategy** section, a committee of models collectively decides on the distributions for synthetic data, where the voting mechanism considers prior performance and calculates a weighted gradient update based on each model's distribution and prediction. **Batch-Specific Soft Labeling** generates soft labels by embedding batch norm statistics from synthetic data batch to mitigate the impact of distribution shift. Finally, a **Smoothed Learning Rate** strategy is applied to the post-training process, adjusting dynamically with a cosine schedule to stabilize training.

ages multiple models' perspectives to create a high-quality, balanced distilled dataset. Voting mechanisms have been widely adopted in NLP and LLM tasks (Yang et al., 2024; Qorib & Ng, 2023; Wang & Plank, 2023). Extending these successes **for the first time** to dataset distillation in computer vision, **CV-DD** leverages a committee-based voting strategy to enhance the diversity and quality of distilled datasets. Our first contribution in this work is to identify pitfalls, disenchant design choices in recent advances on dataset distillation and train an exceptionally strong baseline framework which already achieves state-of-the-art performance. By using a committee of models with different architectures and training strategies, **CV-DD** further enables the capture of a more comprehensive features, enhancing the robustness of the distilled dataset with better accuracy.

Specifically, the proposed **CV-DD** framework introduces a Prior Performance Guided Voting Mechanism, which aggregates distributions and predictions from multiple models to identify representative data points. This approach reduces model-specific biases, promotes diversity in the distilled dataset, and mitigates overfitting by leveraging the unique strengths of each model. Additionally, **CV-DD** enables fine-grained control over the distillation process through our dynamic voting design, where model weights and voting thresholds can be adjusted to prioritize specific features or dataset attributes. We further propose a Batch-Specific Soft Labeling (BSSL) to mitigate the distribution shift between the original dataset and the synthetic data for better post-evaluation performance.

Through extensive experiments on benchmark datasets of CIFAR, Tiny-ImageNet, ImageNet-1K and its subsets, we demonstrate that **CV-DD** achieves significant improvements over traditional single/multi-model distillation methods in both accuracy and cross-model generalization. Our results show that datasets distilled for committee voting consistently yield better performance on post-eval tasks, even in low-data or limited-compute scenarios. By harnessing the collective knowledge of multiple models, **CV-DD** provides a robust solution for dataset distillation, highlighting its potential for applications where efficient data usage and computational efficiency are essential. Beyond conventional optimization-based distillation frameworks, we further demonstrate that CV-DD can be seamlessly incorporated into non–training-based methods such as RDED, and that it generalizes well to more challenging scenarios, including synthetic-to-real transfer.

We make the following contributions in this paper:

- We propose a novel framework, *Committee Voting for Dataset Distillation* (**CV-DD**), which integrates multiple model perspectives to synthesize a distilled dataset that encapsulates rich features and produces high-quality soft labels by batch-specific normalization.

- By integrating recent advancements, refining framework design and optimization techniques, we establish a strong baseline within **CV-DD** framework that already achieves state-of-the-art performance in dataset distillation.

- Through experiments across multiple datasets, we demonstrate that **CV-DD** improves generalization, mitigates overfitting, and outperforms prior methods in data-limited scenarios, highlighting its effectiveness as a scalable and reliable solution for dataset distillation.

## 2 RELATED WORK

**Dataset Distillation.** Dataset distillation aims to generate a compact, synthetic dataset that retains essential information from a large dataset. This approach facilitates easier data processing, reduces training time, and achieves performance comparable to training with the full dataset. Existing solutions typically fall into five main categories: 1) *Meta-Model Matching*: This method optimizes for model transferability on distilled data, involving an outer loop for updating synthetic data and an inner loop for training the network. Examples include DD (Wang et al., 2018), KIP (Nguyen et al., 2021), RFAD (Loo et al., 2022), FRePo (Zhou et al., 2022), LinBa (Deng & Russakovsky, 2022), and MDC (He et al., 2024). 2) *Gradient Matching*: This approach performs one-step distance matching between models, focusing on aligning gradients. Methods in this category include DC (Zhao et al., 2020), DSA (Zhao & Bilen, 2021), DCC (Lee et al., 2022), IDC (Kim et al., 2022), and MP (Zhou et al., 2024). 3) *Distribution Matching*: Here, the distribution of original and synthetic data is directly matched through a single-level optimization. Approaches include DM (Zhao & Bilen, 2023), CAFE (Wang et al., 2022), HaBa (Liu et al., 2022), KFS (Lee et al., 2022), DataDAM (Sajedi et al., 2023), FreD Shin et al. (2024), and GUARD (Xue et al., 2024). 4) *Trajectory Matching*: This method matches the weight trajectories of models trained on original and synthetic data over multiple steps. Examples include MTT (Cazenavette et al., 2022), TESLA (Cui et al., 2023), APM (Chen et al., 2023), and DATM (Guo et al., 2024). 5) *Decoupled Optimization with BatchNorm Matching*: SRe$^2$L (Yin et al., 2023) first proposes to decouple the model training and data synthesis for dataset distillation. After that, many decoupled methods have been proposed, such as G-VBSM (Shao et al., 2024a), EDC (Shao et al., 2024b), CDA (Yin & Shen, 2024), LPLD (Xiao & He, 2024), DELT (Shen et al., 2025), and FADRM (Cui et al., 2025).

**Ensemble Multi-Model Dataset Distillation.** Ensemble-based methods in dataset distillation aim to leverage multiple models to enhance distilled data quality and generalization. Thus far, few works have explored this direction, notably the MTT series (Cazenavette et al., 2022; Cui et al., 2023; Du et al., 2023) and G-VBSM (Shao et al., 2024a). MTT leverages a collection of independently trained teacher models on the real dataset, saving their snapshot parameters at each epoch to generate expert trajectories that guide the distillation process. G-VBSM uses a diverse set of local-to-global matching signals derived from multiple backbones and statistical metrics, enabling more precise and effective matching compared to single-model approaches. However, as the diversity of matching models increases, the framework's overall complexity also grows, which can reduce its efficiency. Both MTT and G-VBSM rely on static ensemble configurations and lack adaptive weighting mechanisms to dynamically adjust each model's contribution. Our proposed *Committee Voting* introduces an adaptive mechanism that adjusts model weights based on prior performance, yielding a more effective and robust distilled dataset with improved results.

## 3 APPROACH

### 3.1 PRELIMINARIES

The goal of dataset distillation is to create a compact synthetic dataset that retains essential information from the original dataset. Given a labeled dataset $\mathcal{D} = \{(u_1, v_1), \ldots, (u_{|\mathcal{D}|}, v_{|\mathcal{D}|})\}$, we aim to learn a synthetic dataset $\mathcal{D}_{\text{syn}} = \{(\hat{u}_1, \hat{v}_1), \ldots, (\hat{u}_{|\mathcal{D}_{\text{syn}}|}, \hat{v}_{|\mathcal{D}_{\text{syn}}|})\}$, where $|\mathcal{D}_{\text{syn}}| \ll |\mathcal{D}|$. The objective is to minimize the performance gap between models trained on $\mathcal{D}_{\text{syn}}$ and those trained on $\mathcal{D}$:

$$\sup_{(u,v) \sim \mathcal{D}} \left| \mathcal{L}\left(\Phi_{\xi_{\mathcal{D}}}(u), v\right) - \mathcal{L}\left(\Phi_{\xi_{\mathcal{D}_{\text{syn}}}}(u), v\right) \right| \leq \delta, \tag{1}$$

where $\delta$ is the allowable gap. This leads to the following optimization problem:

$$\underset{\mathcal{D}_{\mathrm{syn}}, |\mathcal{D}_{\mathrm{syn}}|}{\arg\min} \sup_{(u,v)\sim\mathcal{D}} \left| \mathcal{L}\left(\Phi_{\xi_{\mathcal{D}}}(u), v\right) - \mathcal{L}\left(\Phi_{\xi_{\mathcal{D}_{\mathrm{syn}}}}(u), v\right) \right| \tag{2}$$

The goal is to synthesize $\mathcal{D}_{\mathrm{syn}}$ while determining the optimal number of samples per class.

## 3.2 Pitfalls of Latest Methods

**Diversity and bias issues.** SRe$^2$L (Yin et al., 2023) is an optimization-based method that generates distilled data by aligning the Batch Normalization (BN) statistics of synthetic data with those from the training process while simultaneously ensuring the alignment between synthetic data labels and their true labels. The primary limitation of this method is its reliance on a single backbone network for generating distilled data, resulting in limited diversity and increased model-specific bias.

**Informativeness issues.** Existing ensemble based dataset distillation methods, such as G-VBSM (Shao et al., 2024a) and MTT (Cazenavette et al., 2022), operate under the assumption that all pre-trained models contribute equally during the distillation process. This uniform weighting scheme fails to account for the varying informativeness of individual models, resulting in a distilled dataset that lacks sufficient representational richness.

**Suboptimal soft labels.** Prior generative dataset distillation methods (Shao et al., 2024a; Yin et al., 2023; Cazenavette et al., 2022) overlook the distributional shift between synthetic and original images, leading to suboptimal soft labels that hinder model generalization.

## 3.3 Building a Strong Baseline

We first introduce SRe$^2$L++, an improved baseline that achieves *SOTA performance*. Its superiority over SRe$^2$L is illustrated in Fig. 3.

**Real Image Initialization.** SRe$^2$L++ replaces Gaussian noise with real images for initialization, following EDC (Shao et al., 2024b), which showed improved quality at the same cost.

**Data Augmentation.** SRe$^2$L++ improves recovery on small-resolution datasets by incorporating data augmentation, as shown in Fig. 7.

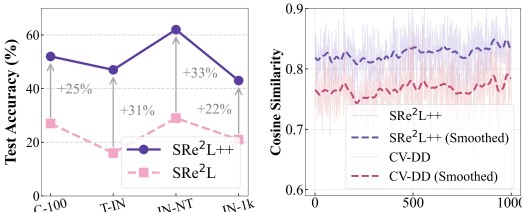

Figure 3: **Left**: Performance gain of strong baseline on four datasets. **Right**: Intra-class cosine similarity on ImageNet-1K (lower is better).

**Batch-Specific Soft Labeling**: To further enhance the performance, we apply the proposed Batch-Specific Soft Labeling technique, detailed in Section 3.6.

**Smoothed Learning Rate and Smaller Batch Size.** Prior studies (Sun et al., 2024) suggest reducing batch size to increase the number of iterations per epoch, thereby mitigating under-convergence, and adopting a smoothed learning rate scheduler to avoid convergence to suboptimal minima. These training settings are applied to training-based methods when compatible. Non–training based methods such as RDED must use larger batches because unoptimized image crops from the original dataset introduce high variance, making BatchNorm unstable with small batches.

---

**Algorithm 1** Prior Performance via Distill-and-Evaluate

---

**Require:** Committee $\mathcal{S}$, dataset $\mathcal{D}$
1: Split $\mathcal{D}$ into $\mathcal{D}_{\mathrm{tr}}$ (80%) and $\mathcal{D}_{\mathrm{ev}}$ (20%).
2: **for** $\Phi \in \mathcal{S}$ **do**
3:     Train $\Phi$ on $\mathcal{D}_{\mathrm{tr}}$ with $\mathcal{L}_{\mathrm{CE}}$; distill $\mathcal{D}_{\Phi}^{\mathrm{dist}}$ using $\Phi$.
4:     Train student $f_{\Phi}$ on $\mathcal{D}_{\Phi}^{\mathrm{dist}}$ with $\mathcal{L}_{\mathrm{KL}} = \mathrm{KL}(\Phi(x) \,\|\, f_{\Phi}(x))$; set $\alpha_{\Phi} \leftarrow \mathrm{Acc}(f_{\Phi}, \mathcal{D}_{\mathrm{ev}})$.
5: **end for**
6: **return** $\{\alpha_{\Phi}\}_{\Phi \in \mathcal{S}}$

---

## 3.4 Overview of **CV-DD**

The overall framework of **CV-DD** is shown in Fig. 2. Built upon the enhanced baseline SRe$^2$L++, **CV-DD** introduces a Prior Performance Guided Voting Strategy that assigns greater influence to stronger models, addressing limitations of prior ensemble methods and improving effectiveness.

### 3.5 Prior Performance Guided Voting Strategy

**Theorem 1** (Proof in Appendix A.1). *Let $\{\Phi_i\}_{i=1}^{|S|}$ be a committee of models, with diversity quantified as $K := \frac{2}{|S|(|S|-1)} \sum_{i<j} \Pr_{x \sim \mathcal{D}}[\Phi_i(x) \neq \Phi_j(x)]$. We denote the gradient at iteration $t$ for sample $\hat{u}_z$ as $G_z^{(t)}$, and suppose: (i) $\|G_z^{(t)}\|^2 \leq G_{\max}$, (ii) $\mathbb{E}[\|G_z^{(t)} - G_{z'}^{(t)}\|^2] \geq C_g K$ for $z, z'$ from the same class and some constant $C_g$. Then the expected cosine distance satisfies:*

$$\mathbb{E}[\Delta_{\cos}^{(t+1)}] \geq \mathbb{E}[\Delta_{\cos}^{(t)}] + \frac{1}{2}\eta^2 C_g K, \tag{3}$$

*where $\Delta_{\cos}^{(t)} := 1 - \cos(\hat{u}_z^{(t)}, \hat{u}_{z'}^{(t)})$, which quantifies the angular discrepancy between normalized update directions. Hence, larger committee diversity leads to increased intra-class separation.*

**Committee Members.** Let $S$ denote the committee comprising $|S|$ diverse backbone architectures, which are selected to introduce architectural diversity, enrich representation capacity, and improve the robustness of the distilled dataset (see Theorem 1 and Fig. 3).

**Prior Performance.** We generate distilled data from pre-trained models, with its quality reflecting the critical information retained. Thus, the generalization of models trained on this data serves as a proxy for the source model's prior performance (detailed in Algorithm 1).

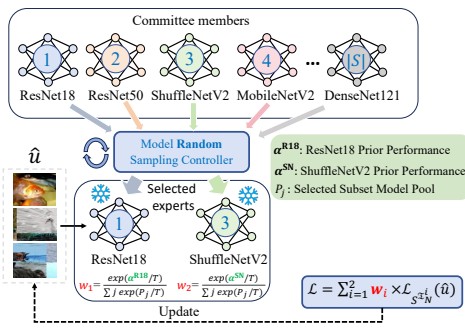

Figure 4: Demonstration of our *Committees Voting*.

**Prior-based Voting in Distilled Data Generation.** See Fig. 4 for details. Let $\mathcal{I}_N \subset \{1, \ldots, |S|\}$ be a randomly sampled subset of $N$ indices, where $2 \leq N \leq |S|$. The $i$-th index in $\mathcal{I}_N$ is denoted by $\mathcal{I}_N^i$, with the corresponding backbone and its prior performance represented as $S^{\mathcal{I}_N^i}$ and $\alpha^{\mathcal{I}_N^i}$, respectively. We define the prior-based voting loss as:

$$\mathcal{L}(\hat{u}) = \sum_{i=1}^{N} \frac{\exp(\alpha^{\mathcal{I}_N^i}/T)}{\sum_{j=1}^{N} \exp(\alpha^{\mathcal{I}_N^j}/T)} \mathcal{L}_{S^{\mathcal{I}_N^i}}(\hat{u}), \tag{4}$$

where SoftMax prioritizes stronger models, steering optimization toward informative directions. In practice, using a small subset (e.g., N=2) balances the influence of the strongest expert while retaining complementary signals, as further analyzed in Section 4.3.

**Prior-based Voting in Soft Label Generation.** Similarly, soft labels are generated through prior-guided voting, where class probabilities from selected models are aggregated via a weighted average determined by their prior performance. This strategy amplifies the influence of informative models, leading to more reliable supervision.

**Theorem 2** (Proof in Appendix A.2). *Let $J(\hat{u})$ denote the generalization risk of the synthetic image. Suppose each model $\Phi_i$ is associated with a prior performance score $\alpha_i > 0$, and contributes a gradient $\nabla_{\hat{u}} \ell(\Phi_i(\hat{u}), \hat{v})$ during optimization. Assume that $\langle \nabla J(\hat{u}), \nabla_{\hat{u}} \ell(\phi_i(\hat{u}), \hat{v}) \rangle = \lambda \cdot \alpha_i, \quad \lambda > 0$. Then, the inner product between the generalization risk gradient and the aggregate update under prior-weighted voting satisfies:*

$$\langle \nabla J(\hat{u}), G_{prior} \rangle > \langle \nabla J(\hat{u}), G_{equal} \rangle, \tag{5}$$

*where $G_{prior}$ and $G_{equal}$ are weighted and uniform gradient averages, with each weight given by a SoftMax over $\alpha_i$.*

The key insight is that prior-guided voting aligns updates more closely with the gradient direction that promotes generalization, compared to uniform averaging.

### 3.6 Batch-Specific Soft Labeling

In the post-evaluation stage, a teacher model is commonly employed to pre-generate soft labels (Shen & Xing, 2022), thereby enhancing the generalization of the student model (Hinton, 2015;

Müller et al., 2019). Typically, the teacher model includes Batch Normalization layers (Sun et al., 2024; Shao et al., 2024a; Yin et al., 2023; Qin et al., 2024), which utilize running statistics to normalize features. These statistics are progressively updated during training, as detailed in Equation 6.

$$\mu_{\text{running}} \leftarrow \lambda \, \mu_{\text{running}} + (1 - \lambda) \, \mu_B, \quad \sigma^2_{\text{running}} \leftarrow \lambda \, \sigma^2_{\text{running}} + (1 - \lambda) \, \sigma^2_B \tag{6}$$

where $\lambda$ is the momentum, and $\mu_B$, $\sigma^2_B$ are the mean and variance of the current batch, respectively. However, as shown in Fig. 5, we observe that even if the generated images match the BN distribution during synthesis, there is still a significant gap between the BN distribution of the synthetic images and that of the original dataset, due to the influence of regularization terms and optimization randomness. To address this, we propose *Batch-Specific Soft Labeling* (BSSL): Instead of using pre-trained BN statistics from the original images in a real dataset, we recompute the BN statistics directly from each batch of synthetic images, keeping all other parameters frozen with the teacher's original pre-trained values each time soft labels are generated. This straightforward adjustment significantly improves the performance of the model during post-training on synthetic data using these soft labels. Given a distilled batch

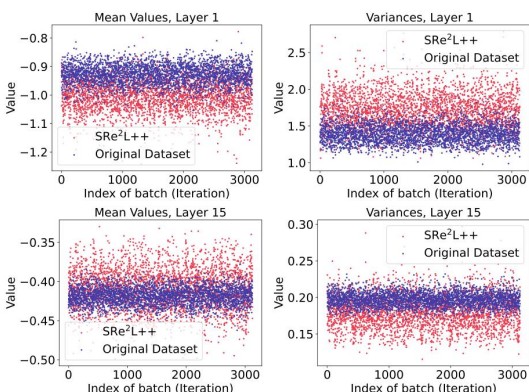

Figure 5: Feature-level statistical discrepancies between synthetic data generated by SRe²L++ and the training data on ImageNet-1K, evaluated across different batches in a pre-trained ResNet18.

$B = \{\hat{u}_i\}_{i=1}^N$ with $\hat{u}_i \in \mathbb{R}^{C \times H \times W}$ denoting the feature map of each sample, the BatchNorm statistics are computed as:

$$\mu_{B,c} = \frac{1}{NHW} \sum_{i,h,w} \hat{u}_{i,c,h,w}, \quad \sigma^2_{B,c} = \frac{1}{NHW} \sum_{i,h,w} (\hat{u}_{i,c,h,w} - \mu_{B,c})^2 + \epsilon$$

where $\mu_{B,c}$ and $\sigma^2_{B,c}$ are the batch statistics for each channel $c$, with a small $\epsilon$ added for numerical stability. This adjustment refines the normalization process with more aligned statistics, thereby enhancing the quality of the soft labels. As shown in Fig. 6, BSSL leads to more aligned embedding statistics across different BN layers.

**Extending BSSL to Non-BN Architectures.** For models without BN layers (ViT), we adapt them to support BSSL by constructing a BN-ViT variant that replaces Layer-Norm with BatchNorm and inserts BN layers in each feed-forward block (Yao et al., 2021). This modification allows non-BN models to benefit from BSSL without affecting their representational capacity, showing that BN reliance does not limit the applicability of CV-DD.

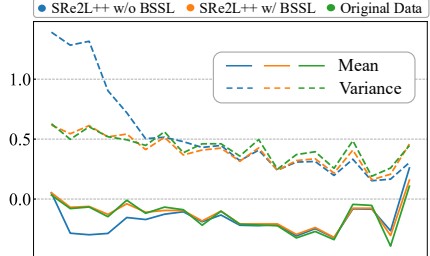

Figure 6: Embedding statistics with and without BSSL across layers.

## 4 EXPERIMENTS

### 4.1 EXPERIMENTAL CONFIGURATION

**Datasets.** To comprehensively evaluate the performance of **CV-DD**, we conduct experiments on both low- and high-resolution datasets. The low-resolution datasets include CIFAR-10/100 (32×32) (Krizhevsky et al., 2009), while the high-resolution datasets comprise Tiny-ImageNet (64×64) (Le & Yang, 2015), ImageNet-1K (224×224) (Deng et al., 2009), and its subsets.

**Baseline Methods.** To evaluate the effectiveness of our tailored ensemble strategy, we include three ensemble-based methods: MTT, G-VBSM, and EDC. Additionally, we consider CDA, RDED and SRe²L++, which incorporate recent advances in dataset distillation.

Table 1: **Comparison with SOTA Baseline Methods.** All models are trained with 300 epochs.

| Dataset | IPC (Ratio) | ResNet18 | | | | ResNet50 | | | | ResNet101 | | | |
|---|---|---|---|---|---|---|---|---|---|---|---|---|---|
| | | CDA | RDED | SRe²L++ | **CV-DD** | CDA | RDED | SRe²L++ | **CV-DD** | CDA | RDED | SRe²L++ | **CV-DD** |
| CIFAR-10 | 1 (0.02%) | - | 22.9 | 24.9 | **27.4** | - | 10.2 | 24.4 | **24.9** | - | 18.7 | 23.4 | **26.6** |
| | 10 (0.2%) | - | 37.1 | 51.3 | **54.7** | - | 33.1 | 47.9 | **49.7** | - | 33.7 | 41.5 | **48.2** |
| | 50 (1.0%) | - | 62.1 | 75.8 | **76.9** | - | 54.2 | 71.4 | **72.1** | - | 51.6 | 73.6 | **74.4** |
| CIFAR-100 | 1 (0.2%) | 13.4 | 11.0 | 12.0 | **16.5** | - | 10.9 | 12.6 | **17.8** | - | 10.8 | 9.9 | **15.1** |
| | 10 (2.0%) | 49.8 | 42.6 | 56.7 | **61.8** | - | 41.6 | 52.1 | **59.6** | - | 41.1 | 54.3 | **61.5** |
| | 50 (10.0%) | 64.4 | 62.6 | 66.6 | **69.4** | - | 64.0 | 67.0 | **70.3** | - | 63.4 | 67.8 | **71.1** |
| Tiny-ImageNet | 1 (0.2%) | 3.3 | 9.7 | 9.3 | **20.1** | - | 8.2 | 7.6 | **17.5** | - | 3.8 | 7.5 | **19.2** |
| | 10 (2.0%) | 43.0 | 41.9 | 46.5 | **53.0** | - | 38.4 | 42.8 | **52.8** | - | 22.9 | 45.4 | **53.9** |
| | 50 (10.0%) | 48.7 | 58.2 | 53.5 | **61.2** | 49.7 | 45.6 | 53.7 | **64.1** | 50.6 | 41.2 | 53.7 | **63.1** |
| ImageNette | 1 (0.1%) | - | 35.8 | 34.7 | **37.5** | - | 27.0 | 25.9 | **29.4** | - | 25.1 | 25.2 | **26.9** |
| | 10 (1.0%) | - | 61.4 | 73.7 | **74.4** | - | 55.0 | 72.6 | **73.9** | - | 54.0 | 66.6 | **67.8** |
| | 50 (5.2%) | - | 80.4 | 80.3 | **81.4** | - | 81.8 | 81.2 | **82.3** | - | 75.0 | 73.4 | **76.3** |
| ImageWoof | 1 (0.1%) | - | 20.8 | 17.0 | **21.3** | - | 17.8 | 14.7 | **22.2** | - | 19.6 | 12.9 | **20.4** |
| | 10 (1.1%) | - | 38.5 | 45.2 | **63.0** | - | 35.2 | 42.3 | **47.5** | - | 31.3 | 38.3 | **45.2** |
| | 50 (5.3%) | - | 68.5 | 64.5 | **68.7** | - | 64.1 | 65.5 | **68.7** | - | 59.1 | 57.8 | **59.6** |
| ImageNet-1k | 1 (0.1%) | - | 6.6 | 8.6 | **12.1** | - | 8.0 | 8.0 | **12.8** | - | 5.9 | 6.2 | **8.4** |
| | 10 (0.8%) | 33.6 | 42.0 | 43.1 | **49.5** | - | 49.7 | 47.3 | **57.0** | - | 48.3 | 51.2 | **57.2** |
| | 50 (3.9%) | 53.5 | 56.5 | 57.6 | **59.5** | 61.3 | 62.0 | 61.8 | **65.3** | 61.6 | 61.2 | 61.0 | **64.6** |

## 4.2 MAIN RESULTS

**High- and Low-Resolution Datasets.** As shown in Table 1, **CV-DD** consistently outperforms state-of-the-art distillation methods across both large- and small-scale benchmarks. On ImageNet-1K at IPC=50 with ResNet-18, **CV-DD** achieves 59.5%, surpassing SRe²L++ by +1.9% and CDA by +6%. Similarly, on CIFAR-100 at IPC=10 with ResNet-18, it reaches 61.8%, outperforming RDED by +19.2%, SRe²L++ by +5.1%, and CDA by +12%. These results highlight the robustness and broad applicability of **CV-DD** across datasets of varying resolutions and scales.

**Comparison with Vanilla Ensemble Methods.** To ensure fair comparison with EDC, **CV-DD** is evaluated under EDC's settings. As shown in Table 2, **CV-DD** consistently outperforms prior vanilla ensemble methods across datasets and resolutions. Notably, at IPC=50, it surpasses EDC by +1.5% on ImageNet-1K, while exceeding MTT by +33.2% on Tiny-ImageNet. These results validate the effectiveness of **CV-DD**'s tailored ensemble approach in distilling high-quality datasets.

Table 2: Comparison between vanilla ensemble methods and our prior-based voting **CV-DD**, using ResNet18 for G-VBSM, EDC, and ours, and Conv128 for MTT on long training setting.

| Dataset | IPC | MTT | G-VBSM | EDC | **CV-DD** |
|---|---|---|---|---|---|
| CIFAR-100 | 10 | 40.1 | 59.5 | 63.7 | **66.0** |
| | 50 | 47.7 | 65.0 | 68.6 | **70.4** |
| Tiny-ImageNet | 10 | 23.2 | - | 51.2 | **53.0** |
| | 50 | 28.0 | 47.6 | 57.2 | **61.2** |
| ImageNet-1k | 10 | - | 31.5 | 48.6 | **49.5** |
| | 50 | - | 51.8 | 58.0 | **59.5** |

## 4.3 ABLATION STUDY

**Impact of $T$ on the Distilled Dataset.** As shown in Table 3a, setting $T = 5$ yields the best overall performance. A low temperature such as $T = 1$ induces excessive sensitivity to score differences and leads to model dominance, while a high temperature like $T = 20$ causes near-uniform weighting across experts, thereby degrading the quality of the distilled dataset.

**Impact of Prior-Based Voting.** To assess the effectiveness of prior-based voting, we conducted ablation studies shown in Table 3b. Results show that prior-based voter consistently outperforms equal and random voters across all configurations, highlighting the advantages of leveraging prior knowledge for high-quality dataset generation.

**Impact of the Number of Experts $N$.** As shown in Table 3c, increasing $N$ to 3 raises computational cost and degrades performance. This is mainly due to **(1) dilution of the strongest recover model's influence**, as its weighting drops from about

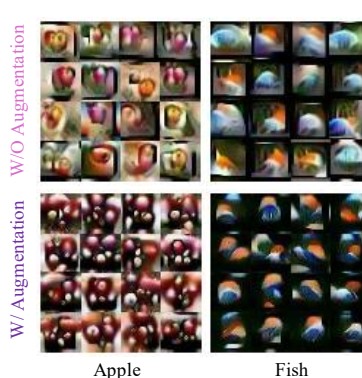

Figure 7: Comparison of distilled data on CIFAR-100 generated by SRe²L++ with and without data augmentation.

Table 3: Ablation experiments. For (a) to (c), The Top row is conducted on CIFAR-100; while the Bottom row is conducted on ImageNet-1K. All results are reported with IPC=10 using ResNet-18.

| $T = 1$ | $T = 5$ | $T = 10$ | $T = 15$ | $T = 20$ |
|---|---|---|---|---|
| 59.3 | **61.8** | 61.3 | 61.0 | 60.9 |
| 45.8 | **49.5** | 49.2 | 49.0 | 48.8 |

(a) Performance across different temperature values ($T$).

| SRe²L++ | CV-DD w/ Random | CV-DD w/ Equal | CV-DD w/ Prior |
|---|---|---|---|
| 56.7 | 59.8 | 60.7 | **61.8** |
| 43.1 | 47.6 | 48.2 | **49.5** |

(b) Comparison of weighting strategies on distilled datasets, where *Equal* assigns uniform weights (0.5) and *Random* samples weights arbitrarily.

| $N = 2$ | $N = 3$ |
|---|---|
| **61.8** | 60.1 |
| **49.5** | 48.7 |

(c) Effect of the number of experts.

| Committee Choices $S$ | | | | | Student Accuracy (%) | |
|---|---|---|---|---|---|---|
| R18 | R50 | D121 | SV2 | MBV2 | CIFAR-100 | ImageNet-1K |
| ✓ | | | | | 56.7 | 43.1 |
| ✓ | ✓ | | | | 60.0 | 43.8 |
| ✓ | ✓ | ✓ | | | 61.0 | 45.4 |
| ✓ | ✓ | ✓ | ✓ | | 61.5 | 48.3 |
| ✓ | ✓ | ✓ | ✓ | ✓ | 61.8 | 49.5 |

(d) Effect of committee size ($S$) on the performance of the student model (ResNet-18) trained with a 10-IPC distilled dataset.

| Method | IPC | CIFAR-100 | | ImageNet-1k | |
|---|---|---|---|---|---|
| | | W/O BSSL | W/ BSSL | W/O BSSL | W/ BSSL |
| SRe²L++ | 1 | 10.5 | **12.0** | 4.2 | **8.6** |
| | 10 | 53.3 | **56.7** | 38.5 | **43.1** |
| **CV-DD** | 1 | 12.9 | **16.5** | 5.8 | **12.1** |
| | 10 | 57.3 | **61.8** | 42.5 | **49.5** |

(e) Performance comparison on CIFAR-100 and ImageNet-1K under varying IPCs, with and without BSSL.

0.9 to 0.5–0.6, and **(2) gradient divergence**, where gradients become less aligned with the best model, reducing dataset fidelity. Hence, $N = 2$ provides the best balance.

**Impact of Committee Size $|S|$.** Table 3d shows that larger committees consistently improve distilled dataset quality, highlighting the benefit of diverse expertise.

**Impact of BSSL.** Table 3e shows that BSSL substantially improves performance under distribution shift between synthetic and real data, achieving gains of +4.6% for SRe²L++ and +7.0% for **CV-DD** at IPC=10 on ImageNet-1K. These results highlight BSSL's ability to provide more reliable supervision in mismatched distributions.

Table 4: Efficiency comparison of ensemble-based methods. Time per image per iteration is measured on an RTX-4090 with batch size 100 using identical committee models. N/A indicates methods that are not scalable.

| Datasets | MTT | G-VBSM | EDC | **CV-DD** |
|---|---|---|---|---|
| ImageNet-1k | N/A | 4.32 ms | 4.99 ms | **1.91** ms |

## 4.4 ANALYSIS

**Overfitting Analysis. CV-DD** effectively mitigates overfitting in the post-training phase. As shown in Fig. 8, it maintains lower training accuracy while consistently achieving higher test accuracy than SRe²L++, indicating better generalization. This underscores the regularization effect of the Prior-Performance-Guided Voting in overfitting-prone scenarios.

**Efficiency Analysis.** We evaluate the efficiency of **CV-DD** against state-of-the-art ensemble methods on ImageNet-1K (Table 4). MTT shows the highest computational cost and scales poorly. In comparison, **CV-DD** is 2.41 ms faster per

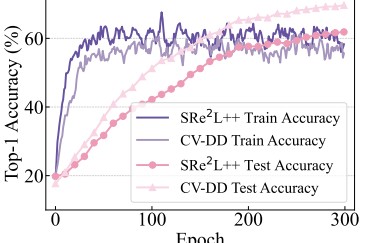

Figure 8: **CV-DD** vs. SRe²L++.

iteration than G-VBSM, which incurs extra overhead from aligning convolutional statistics, similar to EDC. These results demonstrate the clear computational advantage of **CV-DD** over existing ensemble-based approaches. Furthermore, we evaluate the total prior-evaluation cost, which includes pretraining the committee members, generating the distilled data, and running the evaluation. As shown in Table 5a, the full prior-evaluation stage requires 84.7 hours on ImageNet-1k, leading to a total of 137.5 hours for distilling a 50-IPC dataset. In comparison, G-VBSM requires 187.5

Table 5: Detailed prior-evaluation time and total runtime (including prior evaluation) for distillation.

| Model | Pretraining | Generation | Evaluation | Peak GPU Usage | Total Time |
|---|---|---|---|---|---|
| ResNet-18 | 10.1 h | 1.3 h | 1.5 h | 11.3 GB | 12.9 h |
| ResNet-50 | 14.4 h | 4.4 h | 1.5 h | 23.0 GB | 20.3 h |
| ShuffleNetV2 | 9.4 h | 2.6 h | 1.5 h | 7.6 GB | 13.4 h |
| MobileNet-V2 | 10.2 h | 2.4 h | 1.5 h | 11.4 GB | 14.1 h |
| DenseNet-121 | 16.4 h | 6.0 h | 1.5 h | 19.5 GB | 23.9 h |

(a) Detailed prior evaluation time.

| | G-VBSM (hrs) | CV-DD (hrs) |
|---|---|---|
| IPC = 50 | 187.5 | 137.5 |
| IPC = 100 | 375.0 | 190.3 |
| IPC = 150 | 562.5 | 243.1 |
| IPC = 200 | 750.0 | 295.9 |

(b) Total distillation time.

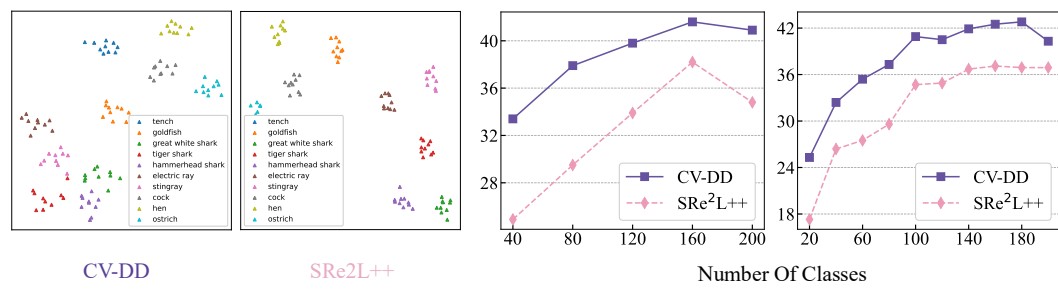

Figure 9: **Left**: t-SNE visualization on ImageNet-1K (IPC=10) for ten selected classes. **Right**: Continual learning results on Tiny-ImageNet (IPC=50) with 5 and 10 steps.

hours, as reported in Table 5b. Notably, once prior evaluation is completed, our method can reuse the recorded model performances for subsequent generations on the same dataset, allowing substantially faster synthesis in future runs.

**Cross-Framework Generalization.** The committee mechanism can be seamlessly integrated into various dataset distillation frameworks. To validate this flexibility, we incorporate it into the non-training-based RDED, where committee voting yields consistent performance gains (Table 6). This demonstrates that CV-DD functions as a versatile plug-in module, enhancing diverse distillation paradigms through model diversity and prior-guided voting.

Table 6: Comparison of RDED, equal voting, and prior voting.

|  | RDED | w/ Equal Voting | w/ Prior Voting (CV-DD) |
|---|---|---|---|
| **IPC = 10** | 42.0 | 43.2 | **44.8** |
| **IPC = 20** | 47.9 | 48.5 | **49.7** |
| **IPC = 30** | 51.7 | 52.1 | **53.2** |

**Cross-Architecture Generalization.** We evaluate **CV-DD** against RDED, EDC, and SRe$^2$L++ across nine diverse architectures, covering lightweight CNNs, large-scale residual networks, and transformer-based models. As shown in Table 7, **CV-DD** achieves the best accuracy on all architectures, demonstrating consistent robustness and strong cross-model generalization. Overall, these results confirm that **CV-DD** generalizes reliably across a wide range of architectures.

Table 7: Top-1 accuracy on ImageNet-1K for cross-architecture generalization with IPC=10.

| Model | #Params | RDED | EDC | SRe$^2$L++ | **CV-DD** |
|---|---|---|---|---|---|
| ShuffleNetV2 | 1.4M | 19.6 | 29.8 | 22.9 | **30.6** |
| MobileNetV2 | 3.4M | 34.4 | 45.0 | 37.1 | **45.6** |
| DenseNet121 | 8.0M | 49.4 | - | 46.7 | **54.7** |
| ResNet18 | 11.7M | 42.0 | 48.6 | 43.1 | **49.5** |
| ResNet50 | 25.6M | 49.7 | 54.1 | 47.3 | **57.0** |
| Swin-Tiny | 28.0M | 29.2 | 38.3 | 28.3 | **39.2** |
| ResNet101 | 44.5M | 48.3 | 51.7 | 51.2 | **57.2** |
| RegNetX-8gf | 39.6M | 51.9 | - | 53.4 | **60.9** |
| WRN-50-2 | 68.9M | 50.0 | - | 50.2 | **58.3** |

**Robustness to Overfitted Teachers.** Overfitted teachers introduce unstable signals that can harm distillation. CV-DD mitigates this through prior-guided voting, which assigns them negligible weights. As shown in Table 8, replacing a standard MobileNetV2 with an overfitted one yields only a minor drop, confirming CV-DD's robustness to noisy teachers.

Table 8: Impact of replacing a standard MobileNetV2 with an overfitted variant in the committee (CIFAR-10, IPC=10).

|  | w/ Original MobileNetV2 | w/ Overfitted MobileNetV2 |
|---|---|---|
| Acc (%) | 54.7 | 54.3 |

**Incorporating ViT in BSSL.** To evaluate the BN-ViT adaptation, we compare BN-ViT with and without BSSL under CV-DD at IPC=10. As shown in Table 9, BN-ViT+BSSL achieves higher accuracy, indicating that BSSL extends effectively to ViT architectures. This confirms that the reliance on BN does not limit the applicability of BSSL across diverse model families.

Table 9: Performance comparison between BN-ViT with and without BSSL.

| ViT (w/o BSSL) | ViT (w/ BSSL) |
|---|---|
| 16.5 | 18.4 |

## 4.5 APPLICATION: CONTINUAL LEARNING

The generalization ability of distilled data in continual learning is a crucial indicator of a distillation method's effectiveness. Following the class-incremental protocol used in DM (Zhao & Bilen, 2023), we evaluate **CV-DD** in a challenging continual learning setting, where the model must sequentially acquire new classes while retaining previously learned ones. As illustrated in Fig. 9, **CV-DD** consistently outperforms SRe$^2$L++, demonstrating better preservation of class separability and reduced

forgetting across increments. These results further underscore the robustness of our distilled data and its strong capacity to support learning under distributional shifts and evolving task structures.

### 4.6 GENERALIZATION TO SYNTHETIC-TO-REAL TRANSFER TASKS

To further assess whether CV-DD generalizes beyond standard vision classification tasks, we evaluate it on a synthetic-to-real transfer benchmark. Specifically, we adopt the VisDA-2017 (Peng et al., 2017) dataset, where the source domain contains purely synthetic images and the target domain consists of real images. This setting directly measures a model's ability to handle substantial distribution shifts. Following the same CV-DD pipeline, we distill an IPC=10 dataset using both CV-DD and SRe$^2$L++. We then train a classifier on each distilled dataset and evaluate it on the real-domain validation set, reporting top-1 accuracy. As shown in Table 10, CV-DD achieves a +1.8 % improvement over SRe$^2$L++, demonstrating its effectiveness in synthetic-to-real transfer.

Table 10: Synthetic-to-real transfer results on VisDA-2017 (top-1 accuracy).

|  | SRe$^2$L++ | CV-DD |
|---|---|---|
| IPC = 10 | 18.9 | **20.7** |

### 4.7 VISUALIZATION

**Effect of Data Augmentation.** As shown in Fig. 7, data augmentation enhances target saliency and information density, leading to improved quality of the distilled dataset.

**t-SNE Visualization.** We extract feature embeddings from a pretrained ResNet-18 for t-SNE (Van der Maaten & Hinton, 2008) visualization (Fig. 9). The SRe$^2$L++ dataset shows compact, well-clustered distributions, indicating limited diversity. In contrast, **CV-DD** produces more dispersed features, reflecting greater variability and better representational richness.

**Distilled Data Visualization.** Fig. 10 compares distilled data from G-VBSM and **CV-DD** on CIFAR-100 and ImageNet-1K. G-VBSM shows lower overall information density, with ImageNet-1K samples often resembling noise. In contrast, **CV-DD** captures more primary visual features, especially on CIFAR-100 with augmentation, yielding more informative synthetic samples.

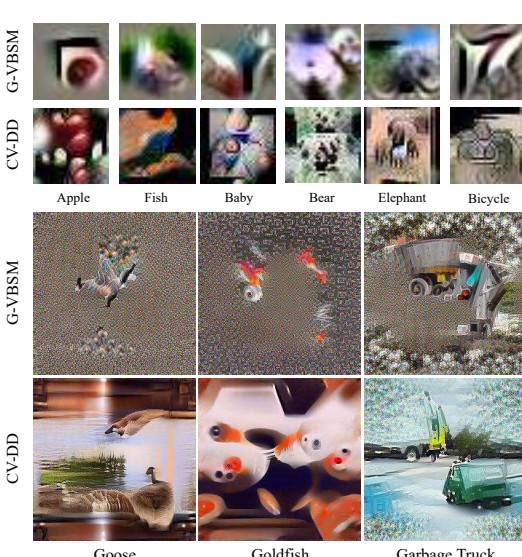

Figure 10: Top two rows: CIFAR-100; Bottom tow rows: ImageNet-1K.

## 5 CONCLUSION

We proposed *Committee Voting* for dataset distillation (**CV-DD**), a novel framework that synthesizes high-quality distilled datasets by leveraging prior performance guided voting strategy for image generation and batch-specific soft labeling for high-quality supervisions. Our approach first establishes a strong baseline that achieves state-of-the-art accuracy through recent advancements and carefully optimized framework design. By combining the distributions and predictions from a committee of models, our method captures rich data features, reduces model-specific biases, and enhances generalization. Complementing this, the generation of high-quality soft labels provides precise supervisory signals, effectively mitigating distribution shifts. Building on these strengths, **CV-DD** consistently improves performance across various configurations and datasets. Our future work will focus on applying the idea of *Committee Voting* to more modalities and applications of dataset distillation tasks.

## ETHICS STATEMENT

We use only public datasets (ImageNet-1K, Tiny-ImageNet) under their licenses; no new human-subject data or sensitive attributes are collected. Before sharing any synthetic samples, we run nearest-neighbor and perceptual-similarity audits to mitigate memorization and exclude near-copies; released artifacts include research-only, lawful-use terms prohibiting surveillance or identification uses. The authors declare no competing interests and adhere to the ICLR Code of Ethics.

## REPRODUCIBILITY STATEMENT

All experiments are conducted on publicly available datasets, CIFAR-10, CIFAR-100, Tiny-ImageNet, and ImageNet-1K. To ensure reproducibility, we fix random seeds for all stochastic components and document all hyperparameters, training schedules, and model architectures in Appendix G. We will release the full codebase along with all scripts for preprocessing, training, and evaluation.

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

# Appendix of CV-DD

CONTENTS

# A  THEORETICAL ANALYSIS

## A.1  COMMITTEE DIVERSITY ENHANCES DATA DIVERSITY

*Proof of Theorem 1.*  We analyze the effect of committee diversity on intra-class separation via the dynamics of gradient-based updates to synthetic images.

Let $\hat{u}_z^{(t)} \in \mathbb{R}^d$ denote the image for sample $z$ at iteration $t$, with update rule:

$$\hat{u}_z^{(t+1)} = \hat{u}_z^{(t)} - \eta G_z^{(t)},$$

where $G_z^{(t)} = \sum_i w_i \nabla_{\hat{u}_z} \ell(\Phi_i(\hat{u}_z), \hat{v}_z)$ is the weighted ensemble gradient from the committee $\{\Phi_i\}$. We assume all samples are **unit-normalized**, i.e., $\|\hat{u}_z^{(t)}\| = 1$.

Let $d^{(t)} := \hat{u}_z^{(t)} - \hat{u}_{z'}^{(t)}$ denote the difference between two intra-class samples. Then,

$$
\begin{aligned}
d^{(t+1)} &= \hat{u}_z^{(t+1)} - \hat{u}_{z'}^{(t+1)} \\
&= \hat{u}_z^{(t)} - \eta G_z^{(t)} - \left( \hat{u}_{z'}^{(t)} - \eta G_{z'}^{(t)} \right) \\
&= d^{(t)} - \eta (G_z^{(t)} - G_{z'}^{(t)}).
\end{aligned}
$$

Taking the squared norm, we have:

$$
\begin{aligned}
\|d^{(t+1)}\|^2 &= \|d^{(t)} - \eta(G_z^{(t)} - G_{z'}^{(t)})\|^2 \\
&= \|d^{(t)}\|^2 + \eta^2 \|G_z^{(t)} - G_{z'}^{(t)}\|^2 \\
&\quad - 2\eta \langle d^{(t)}, G_z^{(t)} - G_{z'}^{(t)} \rangle.
\end{aligned}
$$

Now taking expectation over intra-class pairs $(z, z')$, we obtain:

$$
\begin{aligned}
\mathbb{E}[\|d^{(t+1)}\|^2] &= \mathbb{E}[\|d^{(t)}\|^2] + \eta^2 \mathbb{E}[\|G_z^{(t)} - G_{z'}^{(t)}\|^2] \\
&\quad - 2\eta \mathbb{E}[\langle d^{(t)}, G_z^{(t)} - G_{z'}^{(t)} \rangle].
\end{aligned}
$$

We conservatively drop the final inner product term by assuming that the sample difference direction is uncorrelated with the gradient difference direction, and proceed using Assumption (ii):

$$\mathbb{E}[\|d^{(t+1)}\|^2] \geq \mathbb{E}[\|d^{(t)}\|^2] + \eta^2 C_g K.$$

Now, we relate this result to cosine distance. Under the unit norm assumption, we have:

$$
\begin{aligned}
\|\hat{u}_z^{(t)} - \hat{u}_{z'}^{(t)}\|^2 &= \|\hat{u}_z^{(t)}\|^2 + \|\hat{u}_{z'}^{(t)}\|^2 - 2\langle \hat{u}_z^{(t)}, \hat{u}_{z'}^{(t)} \rangle \\
&= 2(1 - \langle \hat{u}_z^{(t)}, \hat{u}_{z'}^{(t)} \rangle) \\
&= 2\Delta_{\cos}^{(t)}.
\end{aligned}
$$

Thus, $\Delta_{\cos}^{(t)} = \frac{1}{2} \|d^{(t)}\|^2$, and we conclude:

$$
\begin{aligned}
\mathbb{E}[\Delta_{\cos}^{(t+1)}] &= \frac{1}{2} \mathbb{E}[\|d^{(t+1)}\|^2] \\
&\geq \frac{1}{2} \left( \mathbb{E}[\|d^{(t)}\|^2] + \eta^2 C_g K \right) \\
&= \mathbb{E}[\Delta_{\cos}^{(t)}] + \frac{1}{2} \eta^2 C_g K.
\end{aligned}
$$

This completes the proof: committee diversity $K$ induces larger gradient differences, which in turn cause intra-class images to spread out in angular space, increasing $\Delta_{\cos}^{(t)}$.

$\square$

## A.2 PRIOR VOTING ENHANCES DISTILLED DATA QUALITY

*Proof of Theorem 2.* Let $J(\hat{u})$ denote the generalization risk of a synthetic image $\hat{u} \in \mathbb{R}^d$, and let $\nabla J(\hat{u})$ be its gradient. Suppose we have $|S|$ models in the committee, where each model $\Phi_i \in S$ is associated with a known prior performance score $\alpha_i > 0$.

Each model $\Phi_i$ contributes a gradient:

$$g_i := \nabla_{\hat{u}} \ell_{\Phi_i}(\hat{u}),$$

where $\ell_{\Phi_i}(\hat{u})$ denotes the loss incurred by model $\Phi_i$ on synthetic input $\hat{u}$. We assume that the alignment between each model's gradient and the generalization gradient is given by:

$$\langle \nabla J(\hat{u}), g_i \rangle = \lambda \cdot \alpha_i, \quad \text{for all } i \in \{1, \ldots, |S|\},$$

where $\lambda > 0$ is a constant.

We consider two aggregation strategies:

**(1) Equal voting:**

$$G_{\text{equal}} := \frac{1}{|S|} \sum_{i=1}^{|S|} g_i.$$

**(2) Prior-weighted voting using softmax over $\alpha_i$:**

$$w_i := \frac{\exp(\alpha_i)}{\sum_{j=1}^{|S|} \exp(\alpha_j)}, \quad G_{\text{prior}} := \sum_{i=1}^{|S|} w_i g_i.$$

We now compute the alignment between each aggregate direction and the generalization gradient.

**Step 1: Compute** $\langle \nabla J(\hat{u}), G_{\text{equal}} \rangle$

$$
\begin{aligned}
\langle \nabla J(\hat{u}), G_{\text{equal}} \rangle &= \left\langle \nabla J(\hat{u}), \frac{1}{|S|} \sum_{i=1}^{|S|} g_i \right\rangle \\
&= \frac{1}{|S|} \sum_{i=1}^{|S|} \langle \nabla J(\hat{u}), g_i \rangle \\
&= \frac{1}{|S|} \sum_{i=1}^{|S|} \lambda \alpha_i \\
&= \lambda \cdot \bar{\alpha},
\end{aligned}
$$

where $\bar{\alpha} := \frac{1}{|S|} \sum_{i=1}^{|S|} \alpha_i$ is the uniform average of prior performance.

**Step 2: Compute** $\langle \nabla J(\hat{u}), G_{\text{prior}} \rangle$

$$
\begin{aligned}
\langle \nabla J(\hat{u}), G_{\text{prior}} \rangle &= \left\langle \nabla J(\hat{u}), \sum_{i=1}^{|S|} w_i g_i \right\rangle \\
&= \sum_{i=1}^{|S|} w_i \langle \nabla J(\hat{u}), g_i \rangle \\
&= \sum_{i=1}^{|S|} w_i \cdot \lambda \alpha_i \\
&= \lambda \cdot \sum_{i=1}^{|S|} w_i \alpha_i.
\end{aligned}
$$

Let us denote the softmax-weighted average as:

$$\mathbb{E}_w[\alpha] := \sum_{i=1}^{|S|} w_i \alpha_i,$$

then we have:

$$\langle \nabla J(\hat{u}), G_{\text{prior}} \rangle = \lambda \cdot \mathbb{E}_w[\alpha].$$

**Step 3: Show** $\mathbb{E}_w[\alpha] > \bar{\alpha}$

Since $\{w_i\}$ is a softmax distribution over $\{\alpha_i\}$, we can write:

$$w_i = \frac{\exp(\alpha_i)}{\sum_{j=1}^{|S|} \exp(\alpha_j)} \quad \text{for all } i.$$

It assigns greater weight to higher values of $\alpha_i$ in a strictly convex manner. Therefore, unless all $\alpha_i$ are equal (which we exclude), the softmax-weighted average exceeds the arithmetic mean:

$$\mathbb{E}_w[\alpha] > \bar{\alpha}.$$

**Conclusion:**

Combining the results:

$$\langle \nabla J(\hat{u}), G_{\text{prior}} \rangle = \lambda \cdot \mathbb{E}_w[\alpha] > \lambda \cdot \bar{\alpha} = \langle \nabla J(\hat{u}), G_{\text{equal}} \rangle,$$

which shows that prior-weighted voting provides a descent direction that more closely aligns with the gradient of generalization risk, thereby confirming the theorem. $\square$

# B ADDITIONAL ALGORITHMIC DETAILS

## B.1 OPTIMIZATION DETAILS

At each iteration $t$, the optimization aims to update the synthetic image $\hat{u}_z^{(t)}$ by minimizing an objective that jointly enforces prediction alignment and distributional consistency with respect to a fixed pretrained model $\Phi$. The objective is given by:

$$\hat{u}_z^{(t+1)} \leftarrow \arg\min_{\hat{u}} \ \mathcal{L}_{\text{pred}}(\Phi(\hat{u}), \hat{v}_z) + \lambda \mathcal{L}_{\text{BN}}(\hat{u}), \tag{7}$$

where $\mathcal{L}_{\text{pred}}$ ensures semantic alignment with the soft label $\hat{v}_z$, and $\mathcal{L}_{\text{BN}}$ regularizes the feature distribution of $\hat{u}$ to match the population statistics of the original dataset. The coefficient $\lambda$ balances the two objectives.

**Prediction Alignment Loss.** The prediction loss is defined as cross-entropy between the model prediction and the soft label:

$$\mathcal{L}_{\text{pred}}(\Phi(\hat{u}), \hat{v}) = -\sum_{c=1}^{C} \hat{v}_c \log \Phi(\hat{u})_c, \tag{8}$$

where $C$ is the number of classes and $\Phi(\hat{u})_c$ denotes the predicted probability for class $c$.

**BatchNorm Statistic Matching.** To mitigate distributional shift between synthetic and real data, we introduce a regularization term that penalizes discrepancies in BatchNorm statistics:

$$\mathcal{L}_{\text{BN}}(\hat{u}) = \sum_l \left\| \mu_l(\hat{u}) - \text{BN}_l^{\text{RM}} \right\|_2^2 + \sum_l \left\| \sigma_l^2(\hat{u}) - \text{BN}_l^{\text{RV}} \right\|_2^2, \tag{9}$$

where $\mu_l(\hat{u})$ and $\sigma_l^2(\hat{u})$ are the empirical mean and variance of the activations at layer $l$ induced by the synthetic batch $\hat{u}$, and $\text{BN}_l^{\text{RM}}$, $\text{BN}_l^{\text{RV}}$ denote the pretrained running mean and variance recorded on the original dataset. These statistics are frozen and not updated during synthesis. This formulation enforces that synthetic samples activate the network in a manner consistent with the original data distribution, facilitating better generalization under limited supervision.

## B.2 Detailed Voting Process

**Voting in Image Optimization.**

---

**Algorithm 2** Prior Performance Guided Voting Strategy

---

**Require:** Prior Performance $\alpha$, Committee Members $S$, Iteration Budgets $t$
**Ensure:** Synthetic Images $\hat{u}$
1: Initialize Synthetic Images $\hat{u}$
2: $S^{I_2^1}, S^{I_2^2} \leftarrow \text{sample}(S, 2)$;               ▷ Randomly sample two committee members
3: $\alpha^{I_2^1} \leftarrow \alpha[S^{I_2^1}], \alpha^{I_2^2} \leftarrow \alpha[S^{I_2^2}]$;               ▷ Retrieve prior performance scores
4: **for** $i = 1$ to $t$ **do**
5:      Compute weighted loss:

$$\mathcal{L} \leftarrow \sum_{j=1}^{2} \frac{\exp(\alpha^{I_2^j}/T)}{\sum_{k=1}^{2} \exp(\alpha^{I_2^k}/T)} \cdot \mathcal{L}_{S^{I_2^j}}(\hat{u})$$

6:      Perform backpropagation on $\mathcal{L}$
7:      Update synthetic images $\hat{u}$ via gradient descent
8: **end for**
9: Return optimized images $\hat{u}$

---

**Voting in Soft Label Generation.**

---

**Algorithm 3** Prior-Guided Soft Label Aggregation

---

**Require:** Committee $\mathcal{S} = \{\Phi_1, \ldots, \Phi_{|\mathcal{S}|}\}$, prior performance $\{\alpha_i\}_{i=1}^{|\mathcal{S}|}$, temperature $T$, synthetic sample $\hat{u} \in \mathbb{R}^{C \times H \times W}$
**Ensure:** Soft label $\hat{v}$
1: Compute prior-based weights via temperature-scaled SoftMax:

$$w_i \leftarrow \frac{\exp(\alpha_i/T)}{\sum_{j=1}^{|\mathcal{S}|} \exp(\alpha_j/T)} \quad \text{for } i = 1, \ldots, |\mathcal{S}|$$

2: Compute class probabilities from each model:

$$p_i \leftarrow \Phi_i(\hat{u})$$

3: Aggregate weighted soft label:

$$\hat{v} \leftarrow \sum_{i=1}^{|\mathcal{S}|} w_i \cdot p_i$$

4: Return $\hat{v}$

---

## C Additional Experiments

### C.1 Comparison between SRe²L++ and SRe²L

In this section, we present the quantitative improvements of SRe²L++ over SRe²L on both large- and small-scale datasets, as summarized in Table 11. We observe that after incorporating the existing techniques described in Section 3.3 and our proposed BSSL, SRe²L++ achieves substantial performance gains across all settings.

### C.2 Performance Gain from Different Components

In this part, we explicitly quantify the performance gains contributed by each component of the enhanced SRe²L++ baseline. As shown in Table 12, we progressively add individual techniques and

Table 11: Comparison of SRe$^2$L++ with original SRe$^2$L across datasets with ResNet18. Best in each row is **bold**.

| Dataset | IPC (Ratio) | SRe$^2$L | SRe$^2$L++ |
|---|---|---|---|
| CIFAR-100 | 10 (2.0%) | 23.5 | **56.7** |
| | 50 (10.0%) | 51.4 | **66.6** |
| Tiny-ImageNet | 50 (10.0%) | 41.1 | **46.5** |
| ImageNet-1k | 10 (0.8%) | 21.3 | **43.1** |
| | 50 (3.9%) | 46.8 | **57.6** |

Table 12: Ablation study of components contributing to SRe$^2$L++ performance.

| Data Augmentation | Real Data Initialization | Small Batch Size | Smoothed LR | BSSL | Performance |
|---|---|---|---|---|---|
| – | – | – | – | – | 23.3 |
| ✓ | – | – | – | – | 30.8 |
| ✓ | ✓ | – | – | – | 31.3 |
| ✓ | ✓ | ✓ | – | – | 37.6 |
| ✓ | ✓ | ✓ | ✓ | – | 38.5 |
| ✓ | ✓ | ✓ | ✓ | ✓ | 43.1 |

measure their impact. Among all components, the improvement brought by using a small batch size is the most significant.

## C.3 IMPACT OF ONLINE REWEIGHTING

In this part, we introduce an additional baseline that incorporates online reweighting within CV-DD, and compare it against CV-DD with fixed priors. Specifically, under the online reweighting scheme, the two experts start with equal weights of 0.5, and the weights are gradually updated toward their true prior values, reaching the final prior only in the last iteration. We then evaluate the quality of the distilled dataset produced under this strategy. As shown in Table 13 below, the online reweighting approach yields lower performance compared to CV-DD (fixed priors). This degradation occurs because initializing both experts at 0.5 ignores the advantage of the stronger teacher during the early stages of optimization, leading to a suboptimal trajectory and ultimately a lower-quality distilled dataset.

Table 13: Comparison between online reweighting and fixed-prior CV-DD.

| | CV-DD (Online Reweighting) | CV-DD (Fixed Prior) |
|---|---|---|
| Performance | 48.3 | 49.5 |

## C.4 IMPACT OF GRADIENT AGGREGATION

To examine whether CV-DD's can be benefited from partial gradient aggregation, we evaluate a concrete variant of CV-DD where we compute full prediction alignment (cross-entropy) for all experts, but compute the distribution-alignment gradient for only a single expert. This setting reduces the per-iteration computation while retaining part of the supervisory signal used in CV-DD. The per-iteration speedup and the corresponding post-evaluation performance under this partial-gradient setting are reported in Table 14. We observe that, although this approach substantially lowers the computational cost, it also leads to a noticeable degradation in the quality of the distilled dataset. Since CV-DD is already considerably more efficient than existing ensemble-based methods, the additional benefits of adopting partial gradient aggregation remain limited.

Table 14: Comparison between partial and full gradient aggregation.

| | Per-iteration Cost | Performance |
|---|---|---|
| Partial Gradients | 0.96 ms | 47.1 |
| Full Gradients | 1.91 ms | **49.5** |

## D INCORPORATING COMMITTEE VOTING TO RDED

To examine whether the committee-voting idea generalizes to other non-optimization based methods, we further incorporate **CV-DD** into **RDED**, where the original RDED selects patches from the dataset using a single model. Its selection function is defined as:

$$\xi_{i,\star} = \arg \max_{\xi_{i,k} \sim p(\xi_{i,k}|\hat{x}_i)} \left[ -\ell\big(\phi_{\theta_T}(\xi_{i,k}), \phi_h(\xi_{i,k})\big) \right]. \tag{10}$$

We extend this formulation by employing multiple models to select patches and weighting their losses according to each model's prior performance:

$$\xi_{i,\star} = \arg \max_{\xi_{i,k} \sim p(\xi_{i,k}|\hat{x}_i)} \left[ -\sum_{m=1}^{n} w_m\, \ell\big(\phi_{\theta_m}(\xi_{i,k}), \phi_h(\xi_{i,k})\big) \right], \tag{11}$$

where

$$w_m = \frac{\exp(\alpha_m/T)}{\sum_{j=1}^{n} \exp(\alpha_j/T)}. \tag{12}$$

## E  EXPERIMENTAL SETUP

We provide detailed information regarding the hardware, software environment, and reproducibility measures to facilitate faithful replication of our experiments.

- **Hardware.** All experiments were conducted on a dedicated Linux server equipped with 6 NVIDIA RTX 4090 GPUs (each with 24GB memory), 256GB of RAM, and an AMD Ryzen Threadripper PRO 5995WX CPU. The operating system was Ubuntu 22.04.5 LTS. This high-performance computing environment ensured stable and efficient execution of all training and evaluation procedures.

- **Software.** The experiments were implemented using PyTorch version 2.0.1, with CUDA 11.8 for GPU acceleration and Python 3.10 as the primary programming language. Data augmentation pipelines were constructed using the `torchvision` library, and all experiment tracking and logging were managed via the `wandb` platform. All dependencies and library versions are specified in the provided codebase to ensure compatibility and reproducibility.

- **Reproducibility.** To ensure deterministic behavior across runs, we explicitly set random seeds. All hyperparameter settings used in our experiments are documented in Appendix G, and full implementation details, including scripts for data preprocessing, training, and evaluation, are provided in the supplementary material.

- **Experiment Execution.** To ensure the robustness and reliability of our findings, all reported results are averaged over three independent runs. Each run follows the exact same pipeline, including data loading, model initialization, training, and evaluation, under controlled randomization settings. This repeated execution mitigates the impact of stochasticity inherent in deep learning training processes and provides a more stable estimate of performance.

## F  PRIOR PERFORMANCE

**Prior Performance Computation.** This section provides a detailed account of the prior performance of each model across a range of benchmark datasets. We generate a fixed set of 10 synthetic images per class (IPC = 10) for ImageNet-1K and 50 per class (IPC = 50) for other datasets. Each model trained on the distilled images is then evaluated on its corresponding validation set, and the Top-1 classification accuracy is reported. As shown in Table 15, the resulting performance scores serve as a proxy for each model's generalization ability and form the basis for subsequent ensemble weighting and soft label refinement. Importantly, we fix the IPC across all models to ensure that their prior performance scores are within a comparable range. This is particularly critical because the voting temperature parameter $T$ is shared across models (with $T = 5$ by default), and having performance values on a similar scale avoids skewing the voting distribution due to imbalanced score magnitudes.

Table 15: Prior Performance for different models across different datasets.

| Model | CIFAR-10 | CIFAR-100 | Tiny-ImageNet | ImageNette | ImageWoof | ImageNet-1K |
|---|---|---|---|---|---|---|
| ResNet18 | 75.41 | 66.47 | 56.34 | 81.22 | 63.8 | 42.3 |
| ResNet50 | 70.59 | 66.36 | 56.02 | 73.01 | 50.9 | 31.4 |
| DenseNet121 | 75.46 | 65.62 | 56.81 | 77.83 | 63.3 | 34.6 |
| ShuffleNetV2 | 52.63 | 57.22 | 55.41 | 71.47 | 43.1 | 39.3 |
| MobileNetV2 | 71.37 | 65.03 | 56.43 | 72.12 | 53.8 | 33.2 |

# G  HYPERPARAMETER CONFIGURATION

The overall synthetic data generation process adheres to a unified hyperparameter configuration across all experimental settings, as summarized in Table 16. This consistency facilitates fair comparisons and controlled ablation studies. Deviations from this configuration are restricted to two specific stages: (i) the post-evaluation phase, and (ii) the supervised pre-training of committee members. In these stages, hyperparameters are selectively adjusted to accommodate architectural differences in model capacity and the resolution characteristics of the target dataset.

Specifically, during the pre-training phase, we train five distinct architectures on each dataset to construct the committee: ResNet18, ResNet50, ShuffleNetV2, MobileNetV2, and DenseNet121. These models are chosen to span a range of parameter complexities and inductive biases, ensuring both architectural diversity and representational complementarity. Hyperparameters in this phase, such as learning rate schedules and batch sizes, are tuned with respect to each model's convergence behavior and computational profile.

Table 16: Hyperparameters for generating synthetic data across all five datasets.

| **Hyperparameters for Synthetic Data Generation** | |
|---|---|
| Optimizer | Adam |
| Learning rate | 0.25 |
| beta | 0.5, 0.9 |
| epsilon | 1e-8 |
| Batch Size | 100 or 10 (if $C < 100$) |
| Iterations | 2,000 |
| Scheduler | Cosine Annealing |
| Augmentation | RandomResizedCrop, Horizontal Flip |

## G.1  CIFAR-10

This subsection delineates the complete set of hyperparameter configurations employed in our experiments on CIFAR-10, with the goal of facilitating rigorous reproducibility and enabling future benchmarking efforts under consistent conditions.

**Pre-training of Committee Models.** The detailed training configurations for all backbone models utilized in the committee are presented in Table 17. These models are trained on the full CIFAR-10 dataset using standard supervised learning protocols. The selection of hyperparameters, including learning rates, batch sizes, weight decay, and optimization schedules, has been empirically validated to ensure stable convergence and competitive generalization performance across architectures. The consistency of these training settings is critical for maintaining comparability across committee members and ensuring the integrity of the prior-performance estimation process.

**Post-Evaluation Phase.** The hyperparameter settings adopted during the post-evaluation stage on the distilled CIFAR-10 dataset are comprehensively summarized in Table 18. These configurations govern the training of evaluation models on the synthetic data and are designed to reflect standard supervised learning protocols, thereby enabling fair comparisons across distilled datasets. Parameters such as learning rate schedules, number of training epochs, regularization strength, and batch size are carefully selected to ensure stable optimization dynamics while preserving sensitivity to differences in distilled data quality.

Table 17: Hyperparameters for CIFAR-10 Pre-trained Models.

| Hyperparameters for Model Pre-training | |
| --- | --- |
| Optimizer | SGD |
| Learning rate | 0.01 |
| Weight Decay | 1e-4 |
| Batch Size | 64 |
| Epoch | 10 |
| Scheduler | Cosine Annealing |
| Augmentation | RandomResizedCrop, Horizontal Flip |
| Loss Function | Cross-Entropy |

Table 18: Hyperparameters for post-evaluation task on ResNet18, ResNet50 and ResNet101 for CIFAR-10.

| Hyperparameters for Post-Eval on R18, R50 and R101 | |
| --- | --- |
| Optimizer | Adamw |
| Learning Rate | 0.001 (ResNet18, ResNet50) or 0.0005 (ResNet101) |
| Loss Function | KL-Divergence |
| Batch Size | 16 or 10 (if $|S| \leq 16$) |
| Epochs | 300 |
| Scheduler | Cosine Annealing with $\eta = 1$ |
| Augmentation | RandomResizedCrop, Horizontal Flip, CutMix |

## G.2 CIFAR-100

This subsection presents the complete hyperparameter configurations utilized in our CIFAR-100 experiments, offering a transparent and reproducible foundation for future research endeavors.

**Pre-training of Committee Models.** The hyperparameters employed for training the committee models on the original CIFAR-100 dataset are detailed in Table 19. Each model is trained under a consistent supervised learning framework, with dataset-specific adjustments to accommodate the increased complexity and fine-grained nature of CIFAR-100. The configuration includes optimization parameters such as initial learning rate, batch size, learning rate decay schedule, and regularization strength. These settings are empirically tuned to promote convergence stability and to ensure that the resulting models serve as reliable sources for prior-performance estimation within the committee-based distillation framework.

Table 19: Hyperparameters for CIFAR-100 Pre-trained Models.

| Hyperparameters for Model Pre-training | |
| --- | --- |
| Optimizer | SGD |
| Learning rate | 0.1 |
| Weight Decay | 1e-4 |
| Batch Size | 512 |
| Epoch | 200 |
| Scheduler | Cosine Annealing |
| Augmentation | RandomResizedCrop, Horizontal Flip |
| Loss Function | Cross-Entropy |

**Post-Evaluation Phase.** The hyperparameter settings utilized during the post-evaluation stage on the distilled CIFAR-100 dataset are comprehensively summarized in Table 20. These configurations govern the training of evaluation models exclusively on the synthetic data and are selected to ensure a fair and controlled assessment of the generalization capability imparted by the distilled represen-

tations. Parameters such as learning rate, batch size, number of training epochs, and weight decay are carefully chosen to balance convergence speed and evaluation sensitivity, thereby providing a reliable basis for comparative analysis across distillation methods.

Table 20: Hyperparameters for post-evaluation task on ResNet18, ResNet50 and ResNet101 for CIFAR-100.

| Hyperparameters for Post-Eval on R18, R50 and R101 | |
|---|---|
| Optimizer | Adamw |
| Learning Rate | 0.001 (ResNet18, ResNet50) or |
| | 0.0005 (ResNet101) |
| Loss Function | KL-Divergence |
| Batch Size | 16 |
| Epochs | 300 |
| Scheduler | Cosine Annealing with |
| | $\eta = 1$ (ResNet18, IPC=1, 50) or |
| | $\eta = 1$ (ResNet50, IPC=1, 50) or |
| | $\eta = 1$ (ResNet101) or |
| | $\eta = 2$ (ResNet18, IPC=10) or |
| | $\eta = 2$ (ResNet50, IPC=10) |
| Augmentation | RandomResizedCrop, |
| | Horizontal Flip, CutMix |

### G.3 TINY-IMAGENET

**Pre-training of Committee Models.** Table 21 provides a comprehensive summary of the hyperparameter configurations adopted for training the backbone models on the original Tiny-ImageNet dataset. These models serve as committee members in the distillation framework, and their training is conducted under standardized supervised learning protocols. The selected hyperparameters, including optimizer choice, initial learning rate, batch size, weight decay, and learning rate scheduling, are tailored to accommodate the higher input resolution and increased class cardinality of Tiny-ImageNet, thereby ensuring robust convergence and high-quality feature representations for subsequent distillation.

Table 21: Hyperparameters for Training Tiny-ImageNet Pre-trained Models.

| Hyperparameters for Model Pre-training | |
|---|---|
| Optimizer | SGD |
| Learning rate | 0.1 |
| Momentum | 0.9 |
| Batch Size | 128 |
| Epoch | 50 |
| Scheduler | Cosine Annealing |
| Augmentation | RandomResizedCrop, Horizontal Flip |
| Loss Function | Cross-Entropy |

**Post-Evaluation Phase.** The hyperparameter configurations employed during the post-evaluation phase on the distilled Tiny-ImageNet dataset are detailed in Table 22. These settings govern the training of evaluation models exclusively on the synthesized data and are selected to ensure a rigorous and standardized assessment of the distilled set's efficacy. Given the higher resolution and increased class diversity of Tiny-ImageNet, the configurations including learning rate schedule, regularization strategy, and optimization budget are carefully adapted to preserve stability during training while maintaining sufficient sensitivity to distinguish performance across distillation methods.

Table 22: Hyperparameters for post-evaluation task on ResNet18, ResNet50 and ResNet101 for Tiny-ImageNet.

| **Hyperparameters for Post-Eval on R18, R50 and R101** | |
|---|---|
| Optimizer | Adamw |
| Learning Rate | 0.001 (ResNet18) or |
| | 0.001 (ResNet50) or |
| | 0.0005 (ResNet101) |
| Loss Function | KL-Divergence |
| Batch Size | 16 |
| Epochs | 300 |
| Scheduler | Cosine Annealing with |
| | $\eta = 1$ (ResNet18 IPC=50) or |
| | $\eta = 1$ (ResNet50) or |
| | $\eta = 2$ (ResNet18 IPC=1, 10) or |
| | $\eta = 2$ (ResNet101) |
| Augmentation | RandomResizedCrop, |
| | Horizontal Flip, CutMix |

### G.4 SUBSETS OF IMAGENET-1K

**Pre-training of Committee Models.** The full set of training configurations used to pre-train committee models on the ImageNet-1K subsets is summarized in Table 23. These configurations are tailored to accommodate the high-resolution inputs and the fine-grained semantic variability inherent to ImageNet class data. Optimization hyperparameters including learning rate initialization, batch size, weight decay, and scheduler policies are selected to ensure stable convergence across model architectures and to produce reliable feature extractors for use in the distillation process.

Table 23: Hyperparameters for Training subsets of ImageNet-1k Pre-trained Models.

| **Hyperparameters for Model Pre-training** | |
|---|---|
| Optimizer | SGD |
| Learning rate | 0.01 |
| Momentum | 0.9 |
| weight decay | 1e-4 |
| Batch Size | 64 |
| Epoch | 300 |
| Scheduler | Cosine Annealing |
| Augmentation | RandomResizedCrop, Horizontal Flip |
| Loss Function | Cross-Entropy |

**Post-Evaluation Phase.** The hyperparameter configurations employed during the post-evaluation phase on the distilled subsets of the ImageNet-1K dataset are presented in Table 24. These settings define the training protocols for evaluating models exclusively on the synthesized data and are carefully selected to reflect standardized supervised learning procedures. Particular attention is given to the choice of optimization parameters, including learning rate schedules, regularization strategies, and training durations, in order to ensure a fair and sensitive evaluation of the generalization capacity induced by the distilled representations.

### G.5 IMAGENET-1K

This subsection offers a comprehensive account of the hyperparameter configurations adopted in the ImageNet-1K experiments, with the objective of facilitating reproducibility and enabling consistent comparisons in future research. As the generation of distilled data relies on the use of publicly available ImageNet-1K models provided by the official PyTorch repository, no modifications are

Table 24: Hyperparameters for post-evaluation task on ResNet18, ResNet50 and ResNet101 for subsets of ImageNet-1k.

| Hyperparameters for post-eval on R18, R50 and R101 | |
| --- | --- |
| Optimizer | Adamw |
| Learning Rate | 5e-4 (ResNet18) or |
| | 5e-4 (ResNet50, IPC=1, 50) or |
| | 5e-4 (ResNet101, IPC=1, 10) or |
| | 0.001 (ResNet50, IPC=10) or |
| | 0.001 (ResNet101, IPC=50) |
| Loss Function | KL-Divergence |
| Batch Size | 10 |
| Epochs | 300 |
| Scheduler | Cosine Annealing with |
| | $\eta = 1$ (ResNet18, IPC=10) or |
| | $\eta = 1$ (ResNet50, IPC=10, 50) |
| | or |
| | $\eta = 2$ (ResNet18, IPC=1, 50) or |
| | $\eta = 2$ (ResNet50, IPC=1) or |
| | $\eta = 2$ (ResNet101) |
| Augmentation | RandomResizedCrop, |
| | Horizontal Flip, CutMix |

made to the pretraining phase. Consequently, only the hyperparameters associated with the post-evaluation stage are reported, as summarized in Table 25.

Table 25: Hyperparameters for post-evaluation task on ResNet18, ResNet50 and ResNet101 for ImageNet-1K.

| Hyperparameters for Post-Eval on R18, R50 and R101 | |
| --- | --- |
| Optimizer | Adamw |
| Learning Rate | 0.0005(ResNet101, IPC=50) or |
| | 0.001 |
| Loss Function | KL-Divergence |
| Batch Size | 16 |
| Epochs | 300 |
| Scheduler | Cosine Annealing with |
| | $\eta = 1$ (ResNet18, IPC=50) or |
| | $\eta = 1$ (ResNet50, IPC=50) or |
| | $\eta = 2$ (ResNet18, IPC=1, 10) or |
| | $\eta = 2$ (ResNet50, IPC=1, 10) or |
| | $\eta = 2$ (ResNet101) |
| Augmentation | RandomResizedCrop, |
| | Horizontal Flip, CutMix |

## G.6 CROSS-ARCHITECTURE GENERALIZATION

This subsection provides a detailed exposition of the hyperparameter configurations employed in the cross-architecture generalization experiments, with the aim of promoting reproducibility and facilitating robust comparisons in future studies. To accommodate variations in model capacity, the experimental protocol differentiates between architectures based on their parameter scales. Specifically, architectures with relatively large parameter counts, such as DenseNet201, are assigned the same hyperparameter settings as those used for ResNet101. Conversely, lightweight models adopt the configurations associated with ResNet18 and ResNet50. The complete set of hyperparameters utilized across these architectural variants is documented in Table 25.

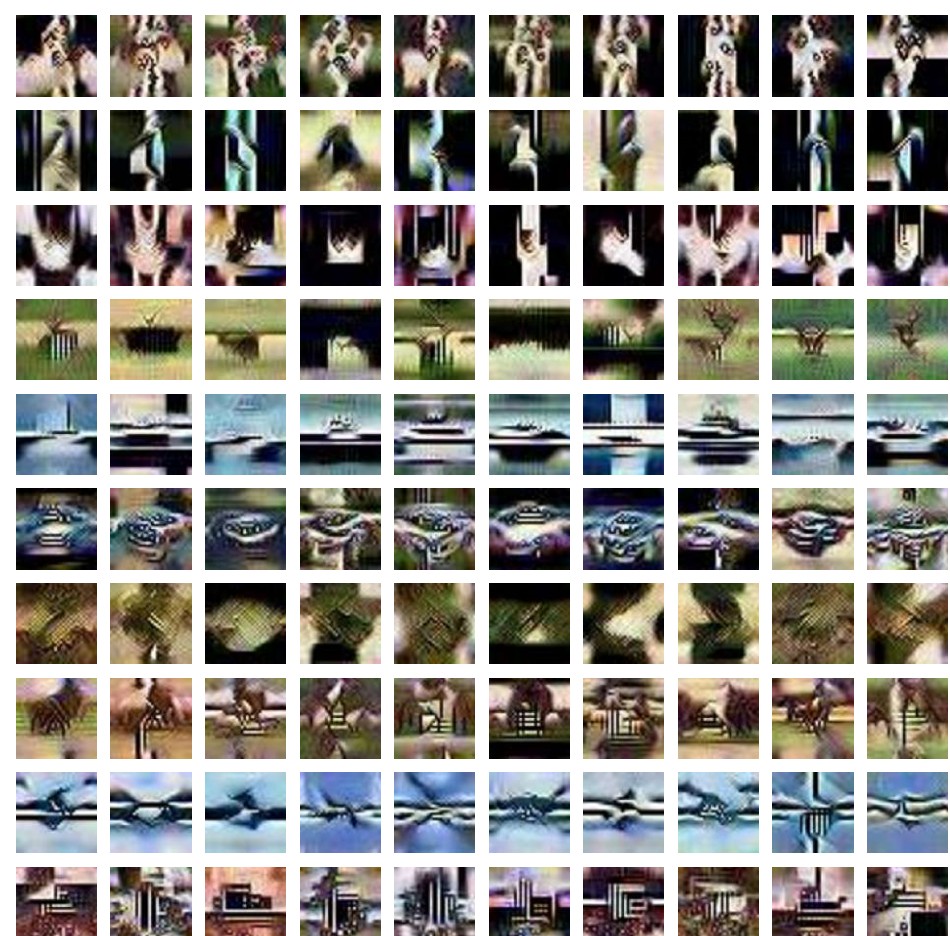

Figure 11: Visualization of synthetic data on CIFAR-10 generated by CV-DD.

## H    DISTILLED DATA VISUALIZATION

Extended qualitative results showcasing the visual characteristics of the distilled data synthesized by
**CV-DD** are provided in Fig. 11 (CIFAR-10), Fig. 12 (CIFAR-100), Fig. 13 (Tiny-ImageNet), Fig. 14
(ImageWoof), Fig. 15 (ImageNette), and Fig. 16 (ImageNet-1K). These visualizations illustrate the
diversity, class discriminability, and structural fidelity of the synthesized samples across datasets of
varying resolution and semantic complexity, thereby offering intuitive evidence of the effectiveness
of the proposed distillation framework.

## I    USE OF LARGE LANGUAGE MODELS

A large language model was used exclusively to improve the writing of the manuscript. All the idea,
method and experimental designs are contributed by the authors.

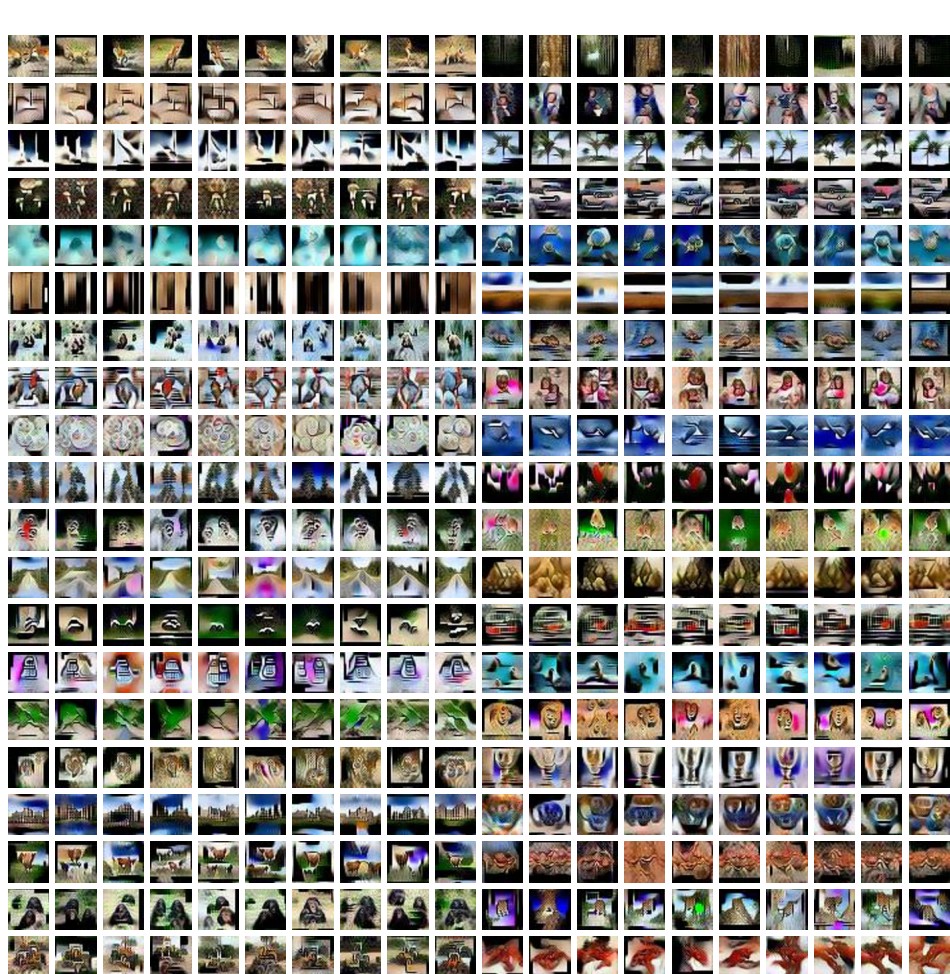

Figure 12: Visualization of synthetic data on CIFAR-100 generated by CV-DD.

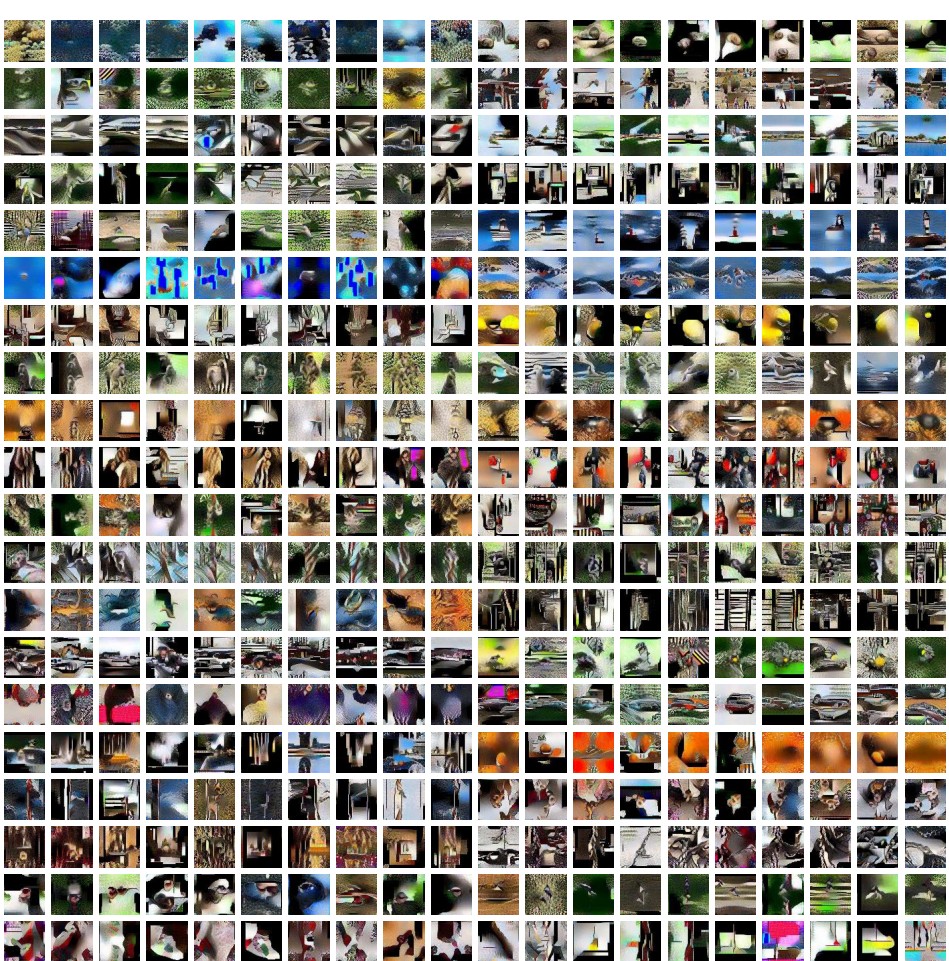

Figure 13: Visualization of synthetic data on Tiny-ImageNet generated by CV-DD.

1512
1513
1514
1515
1516
1517
1518
1519
1520
1521
1522
1523
1524
1525
1526
1527
1528
1529
1530
1531
1532
1533
1534
1535
1536
1537
1538
1539
1540
1541
1542
1543
1544
1545
1546
1547
1548
1549
1550
1551
1552
1553
1554

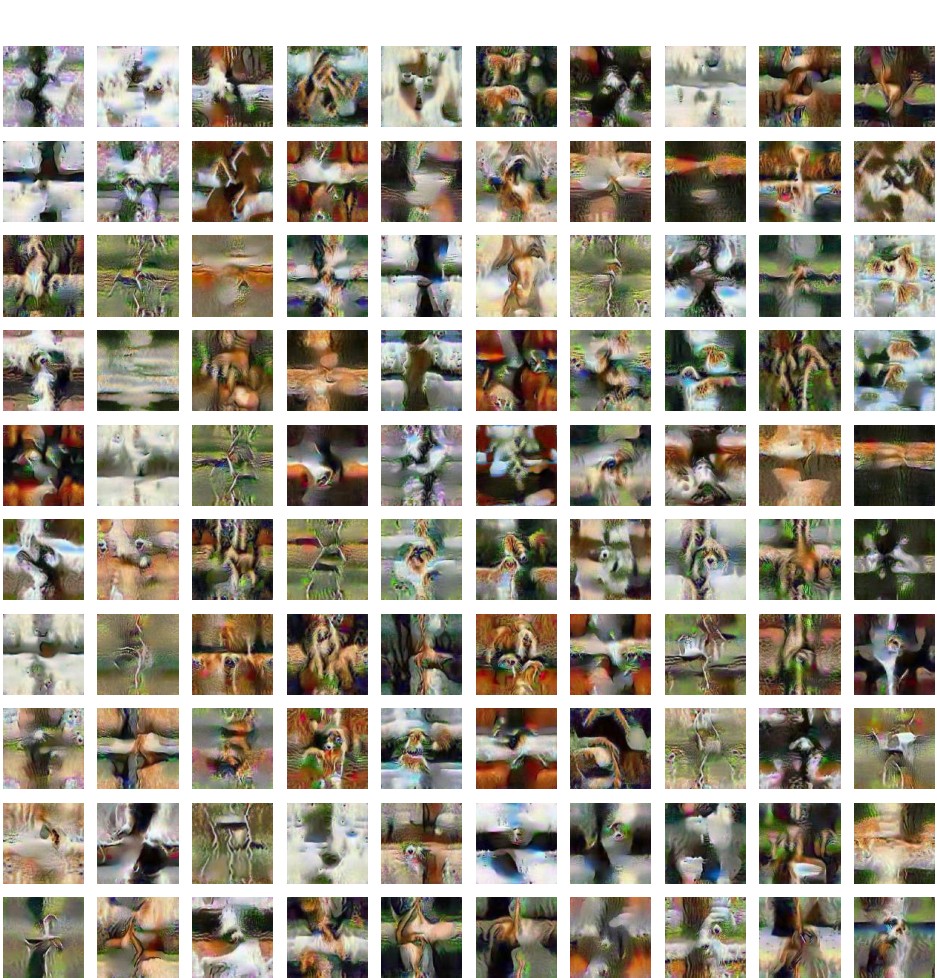

Figure 14: Visualization of synthetic data on ImageWoof generated by CV-DD.

1555
1556
1557
1558
1559
1560
1561
1562
1563
1564
1565

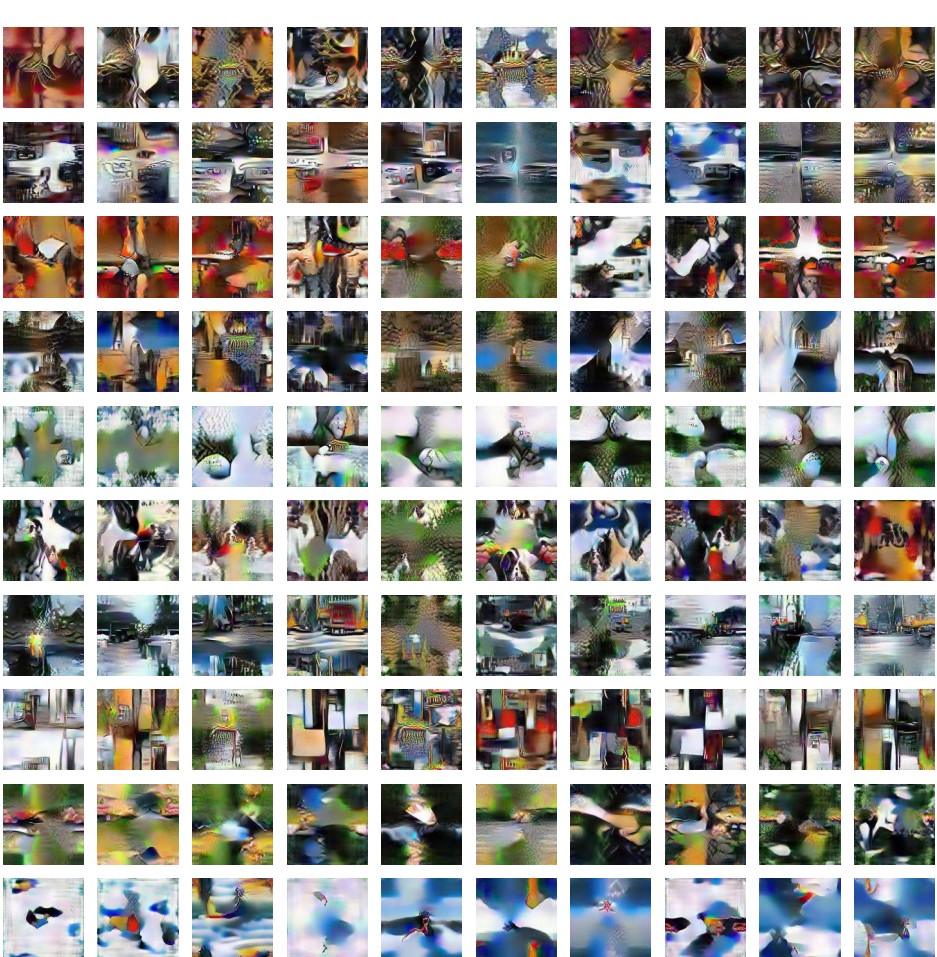

Figure 15: Visualization of synthetic data on ImageNette generated by CV-DD.

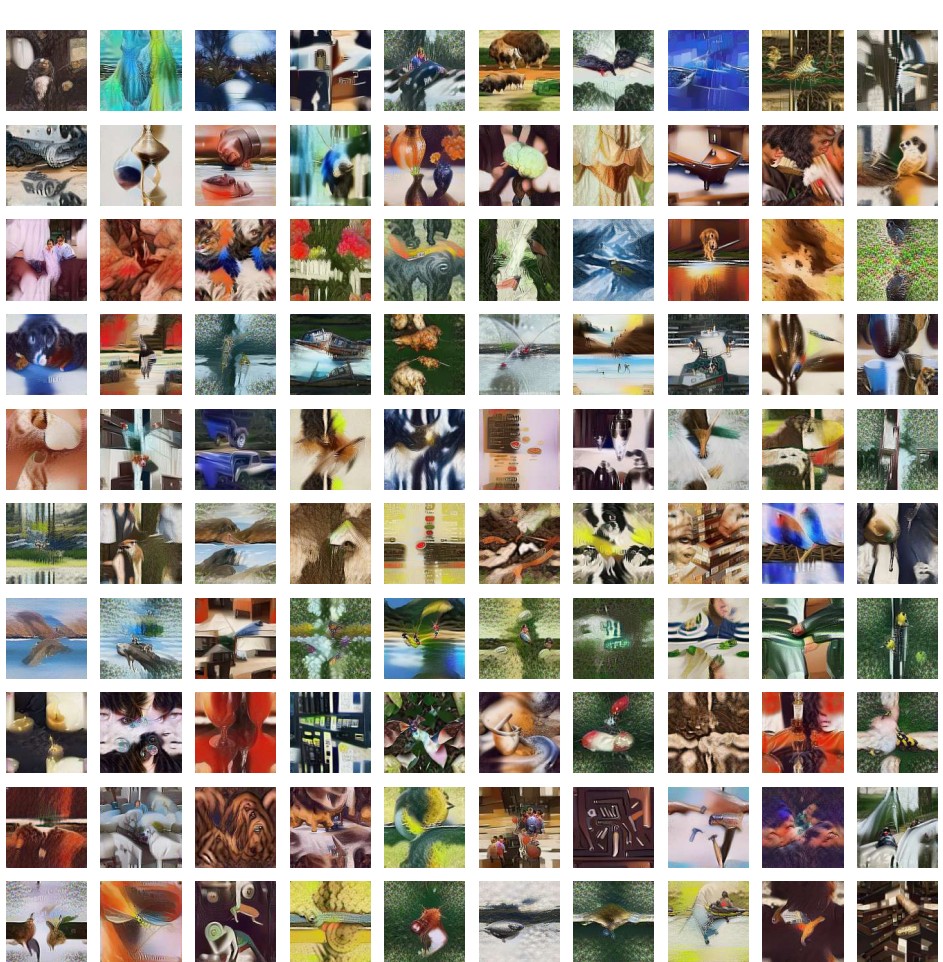

Figure 16: Visualization of synthetic data on ImageNet-1K generated by CV-DD.

