# OpenReview forum: "Dataset Distillation via Committee Voting"
_ICLR.cc/2026/Conference — Submitted to ICLR 2026_

### Official Review · Reviewer_4Bi4 · 2025-10-15

**Soundness:** 3
**Presentation:** 3
**Contribution:** 3
**Rating:** 6
**Confidence:** 3

**Summary:**

This paper proposes CV-DD (Dataset Distillation via Committee Voting), a novel framework that improves dataset distillation (DD) by integrating knowledge from multiple models instead of relying on a single teacher backbone.
The key insight is that existing decoupled DD methods suffer from *single-model bias* and limited diversity, as the synthetic data inherits the inductive bias of one architecture.
CV-DD introduces a committee voting mechanism, where several heterogeneous teacher models (e.g., ResNet18/50, MobileNetV2, ShuffleNetV2, DenseNet121) contribute to the synthesis process.
Each teacher’s contribution is weighted by its *prior performance* (estimated on real data), and gradients are aggregated via a softmax-weighted voting rule.
Theoretical analyses show that model diversity encourages more diverse synthetic data and that prior-guided voting aligns the update direction with better generalization.

**Strengths:**

* **Clear motivation and insight:**
  The paper addresses a genuine and underexplored issue in dataset distillation — the *bias introduced by single-teacher distillation*.
  Framing the solution as a “committee voting” problem is intuitive yet novel.
* **Solid technical design:**
  The combination of prior-guided voting and batch-specific soft labeling is well-justified and effectively integrated into the distillation loop.
  The enhancements (SRe²L++ baseline, real initialization, smoothed learning rate) are reasonable and necessary for a fair comparison.
* **Theoretical support:**
  The paper provides formal analysis showing that committee diversity contributes to broader gradient coverage and improved generalization.
  This strengthens the conceptual grounding of the method.
* **Empirical improvements:**
  Consistent gains across multiple benchmarks, including large-scale ImageNet-1K, demonstrate practical utility and scalability.
  The results are competitive and in several cases surpass existing state-of-the-art methods.

**Weaknesses:**

* **Computation cost and scalability:**
  The committee-based design involves multiple teachers and prior evaluation steps, which may substantially increase computation.
  The paper should report training time, GPU hours, and discuss trade-offs between accuracy and cost more explicitly.
* **Limited domain diversity:**
  The framework is validated only on image classification datasets.
  It would be valuable to explore its applicability to other modalities (e.g., graph or multimodal data) or to synthetic-to-real transfer tasks.

**Questions:**

1. How does CV-DD’s computational cost scale with the number of committee members? Can it be reduced by low-rank or partial gradient aggregation?
2. How robust is the prior-guided voting to noisy or overfitted teacher models? Would dynamic online re-weighting (instead of fixed priors) help?
3. Have you tested CV-DD on other domains (e.g., graph or cross-modal distillation) to confirm generality?

---

> ### Author Response · Authors · 2025-11-29
> **Response to Reviewer 4Bi4 [Part 1/2]**
>
> Thank you for your thoughtful and constructive feedback. We have uploaded a revised version of our paper incorporating all concrete revisions following your suggestions. We would like to further provide detailed point-by-point responses to address your concerns as follows:
>
> >W1. Computation cost and scalability: The committee-based design involves multiple teachers and prior evaluation steps, which may substantially increase computation. The paper should report training time, GPU hours, and discuss trade-offs between accuracy and cost more explicitly.
>
> Thanks for raising this important point. To address it, we first report the total running time of G-VBSM for generating a 50-IPC dataset on a single RTX 4090. G-VBSM requires 4.32 ms per iteration and 4000 iterations to optimize a single image, resulting in approximately 17.3 seconds per image, or about **187.5 hours** in total to generate a 50-IPC dataset. Following the same estimation, our method requires only 52.8 hours for generating a 50-IPC dataset.
>
> We further provide a detailed breakdown of the total time required for Pretraining, Generation, and Evaluation of all committee members on ImageNet-1k, with all experiments conducted on a single RTX-4090. We observed that the original prior performance evaluation setting for ImageNet-1k was not optimal, as generating a 50-IPC dataset solely for evaluation is highly time-consuming. Instead, we found that using a 10-IPC distilled dataset for prior performance evaluation achieves nearly the same quality of distilled data, while substantially reducing the computational cost, as shown in the table below.
>
> |  |  Using **IPC=10** as Prior Performance Evaluation   | Using **IPC=50** as Prior Performance Evaluation |
> |:---:|:---:|:---:|
> |IPC=10| 49.7| 49.5|
> |IPC=50| 59.3 | 59.5|
>
> As shown in the table below (all experiments are conducted using a single RTX-4090), the total computation time for generating an IPC = 50 dataset is 137.5 hours (including 84.7 hours for prior evaluation and 52.8 hours for distillation), whereas G-VBSM requires 187.5 hours under the same setting. This demonstrates that, despite the additional prior evaluation stage, our approach achieves superior efficiency due to its significantly reduced per-iteration cost and fewer total optimization steps per image, while also delivering better performance than state-of-the-art ensemble methods.
>
>
> |     | Pretraining  | Generation | Evaluation | Peak GPU Usage | Total Time |
> |:---|:---:|:----:|:----:|:----:|:----:|
> |   ResNet-18  |   10.1 hours     |   1.3 hours  | 1.5 hours | 11.3 GB | 12.9 hours |
> |   ResNet-50  |   14.4  hours    | 4.4 hours |  1.5 hours   | 23.0 GB| 20.3 hours |
> |   ShuffleNetV2  |   9.4 hours     | 2.6 hours  | 1.5 hours | 7.6 GB | 13.4 hours|
> |   MobileNet-V2  |    10.2 hours    | 2.4 hours   | 1.5 hours| 11.4 GB| 14.1 hours|
> |   Densenet121  |   16.4 hours     |  6 hours   | 1.5 hours|19.5 GB |23.9 hours |
>
> Moreover, we compare the total generation time of G-VBSM across different IPC settings in the table below. As illustrated, this cumulative efficiency allows our framework to become increasingly advantageous as the target IPC grows. For instance, when distilling a 100-IPC dataset on ImageNet-1K, our method requires only 190.3 hours, whereas G-VBSM takes approximately 375 hours on a single RTX-4090. This analysis has been added to **Sec 4.4 (Table 5)** in the revised version.
>
> |    | G-VBSM | CV-DD  |
> |:---|:---:|:----:|
> |   IPC=50  |   187.5 hours    |  137.5 hours|
> |   IPC=100  |  375.0 hours |  190.3 hours|
> |   IPC=150  |  562.5 hours |  243.1 hours|
> |   IPC=200  |  750.0 hours |  295.9 hours|
>
>
>
> >W2. Limited domain diversity: The framework is validated only on image classification datasets. It would be valuable to explore its applicability to other modalities (e.g., graph or multimodal data) or to synthetic-to-real transfer tasks.
>
> Thanks for the insightful suggestion. As recommended, we additionally evaluate CV-DD on a synthetic-to-real transfer benchmark. We adopt the VisDA-2017 dataset [1], where the source domain consists entirely of synthetic images and the target domain comprises real images. This setup directly measures the model’s ability to generalize across significant distribution shifts.
>
> Following the same CV-DD pipeline, we distill an IPC = 10 dataset using both CV-DD and SRe$^2$L++. We then train a classifier on each distilled dataset and evaluate it on the real-domain validation set, reporting top-1 accuracy. The results below show a gain of +1.8 with CV-DD, demonstrating its effectiveness in synthetic-to-real transfer. We have added this experiment in **Sec 4.6**.
>
>
> |     |  SRe$^2$L++   | CV-DD  |
> |:---|:---:|:----:|
> |   IPC=10  | 18.9 | **20.7**   |

---

> ### Author Response · Authors · 2025-11-29
> **Response to Reviewer 4Bi4 [Part 2/2]**
>
> >Q1 (1). How does CV-DD’s computational cost scale with the number of committee members?
>
> Thanks for this insightful question. We would like to clarify that the computational cost of CV-DD **does not scale with the full committee size** $|S|$. At each iteration, only **N experts** are randomly sampled and used for optimization, so the per-iteration cost grows as **O(N)** rather than **O(|S|)**. In practice, we fix $N=2$ as it provides a good balance between leveraging the strongest teacher and incorporating complementary knowledge from weaker ones as shown in the experiment in Table 4. For the reviewer’s convenience, we include the table below.
>
> | Datasets     | N=2  |N=3 |
> |:--------------|:------:|:--------:|
> | CIFAR-100  | 61.8  |60.1 |
> | ImageNet-1k  |  49.5 | 48.7 |
>
> >Q1 (2). Can it be reduced by low-rank or partial gradient aggregation?
>
> Thanks for asking this insightful question. To assess whether CV-DD’s efficiency can be improved via partial gradient aggregation, we evaluate a modified variant of the framework. In this variant, prediction alignment (cross-entropy) is computed for all experts as usual, but the distribution-alignment gradient is computed for only a single expert. This design lowers per-iteration cost while retaining part of CV-DD’s supervisory signal.
>
> The per-iteration speedup and the corresponding post-evaluation performance under this partial-gradient setting are reported in the table below. Although partial aggregation substantially reduces computation, it also introduces a clear drop in distilled-data quality. Given that CV-DD is already considerably more efficient than existing ensemble-based methods, the marginal gains from partial gradient aggregation do not outweigh the performance loss. We have included this experiments in **Appendix C.4**.
>
> |      | Per-iteration Cost | Performance |
> |:--------------|:------:|:--------:|
> | Partial Gradients  | 0.96 ms  | 47.1  |
> | Full Gradients|1.91 ms | **49.5**|
>
>
> >Q2 (1). How robust is the prior-guided voting to noisy or overfitted teacher models?
>
> Thanks for the thoughtful question. CV-DD is inherently robust to noisy or overfitted teachers. Overfitted teachers typically exhibit low prior performance, and our prior-guided voting assigns them exponentially smaller weights through the Softmax weighting mechanism. Consequently, their gradients contribute minimally to the synthetic data optimization, effectively suppressing their negative influence.
>
> As shown in the table below (experiment conducted on CIFAR-10 IPC=10, where the original MobileNetV2 is trained for 10 epochs and the overfitted version is trained for 100 epochs), replacing MobileNetV2 with its overfitted counterpart in the committee leads to only a slight performance drop. This empirically validates the robustness of CV-DD to overfitted teachers. We have included this experiment in **Sec 4.4 (Table 8)**.
>
> | CV-DD w/ Original MobileNetV2 | CV-DD w/ Overfitted MobileNetV2 |
> |:------:|:--------:|
> | 54.7  | 54.3 |
>
> >Q2 (2).  Would dynamic online re-weighting (instead of fixed priors) help?
>
> Thanks for this insightful question. To address this concern, we introduce an additional baseline that incorporates online reweighting within CV-DD, and compare it against CV-DD with fixed priors. Specifically, under the online reweighting scheme, the two experts start with equal weights of 0.5, and the weights are gradually updated toward their true prior values, reaching the final prior only in the last iteration. We then evaluate the quality of the distilled dataset produced under this strategy.
>
> As shown in the table below, the online reweighting approach yields lower performance compared to CV-DD (fixed priors). This degradation occurs because initializing both experts at 0.5 ignores the advantage of the stronger teacher during the early stages of optimization, leading to a suboptimal trajectory and ultimately a lower-quality distilled dataset. We have included this experiment in **Appendix C.3**
>
> | CV-DD (Online  Reweighting) | CV-DD (Fixed Prior) |
> |:------:|:--------:|
> | 48.3 |  49.5 |
>
> >Q3. Have you tested CV-DD on other domains (e.g., graph or cross-modal distillation) to confirm generality?
>
> Thanks for this insightful question. We have evaluated CV-DD on **synthetic-to-real transfer** tasks as you suggested above in our response to Weakness 2. The summarized results are as follows. More details can be referred to our response to Weakness 2.
>
> |     |  SRe$^2$L++   | CV-DD  |
> |:---|:---:|:----:|
> |   IPC=10  | 18.9 | **20.7**   |
>
> --------
> References:
>
> [1] Xingchao Peng, Ben Usman, Neela Kaushik, Judy Hoffman, Dequan Wang and Kate Saenko1. "VisDA: The Visual Domain Adaptation Challenge."

---

### Official Review · Reviewer_kxaT · 2025-10-21

**Soundness:** 2
**Presentation:** 3
**Contribution:** 2
**Rating:** 2
**Confidence:** 4

**Summary:**

This paper proposes CV-DD, a dataset distillation framework that introduces a Prior-based Voting Strategy on top of SRe²L. The authors also incorporate several training refinements, such as real-image initialization, data augmentation, and smoothed learning rate scheduling, claiming SOTA performance under multiple IPC settings. However, the novelty appears marginal, and the fairness of the experiments is questionable.

**Strengths:**

The overall presentation quality is good. The figures are visually clear and help the reader understand the methodology.

The motivation of performance-guided voting is intuitively reasonable in principle.

**Weaknesses:**

The proposed method appears to be a marginal modification over SRe²L. The core contribution, i.e., “Prior-based Voting”, is computationally expensive yet results in only marginal improvements, as shown in Table 4 (middle).

Several enhanced training tricks are adopted only for the proposed method and SRe²L++, like "Smoothed Learning Rate & Smaller Batch Size", which is not applied to other competing methods, leading to potentially unfair comparisons.

The comparison methods are limited. The main results only include two valid baselines, RDED and SRe²L, while another relevant method, CDA, which also follows a similar paradigm as SRe²L, is missing under most experimental settings. In addition, several comparison methods are relatively outdated.

The reported results are inappropriately cited. For instance, the MTT baseline is evaluated under an altered setting that deviates from its official configuration and appears to be intentionally adjusted in favor of the proposed method.

There are typos, e.g., in line 323, “Large scale dataset” should be written in lowercase.

**Questions:**

- What is the architecture used in Fig. 1? Please specify it clearly.

- Regarding the Prior-based Voting in Section 3.4 and Algorithm 1, it appears that distillation and evaluation must be conducted separately for each model in the committee set, which will incur substantial computational overhead. In contrast, SRe2L also relies on pretrained models, but those are standard models that can often be assumed to be readily available.

- Moreover, the MTT results in Table 2 are substantially lower than the original reported performance. What accounts for this degradation? Was it re-implemented using an ensemble strategy? If so, this deviates from the original setting and may distort the comparison, especially since the reported results are significantly lower than those reported in the original MTT paper.

- The ablation study is incomplete and fails to isolate the contribution of key components. Specifically, the impact of the proposed Prior-based Voting is not evaluated independently. Moreover, several additional enhancements, such as Real Image Initialization, Data Augmentation, and Smoothed Learning Rate with Smaller Batch Size, are grouped together without showing their individual effects on performance. If these techniques are treated as default settings, they should be consistently applied to all compared methods (not only to SRe²L) to ensure a fair and unbiased comparison.

- The reported results are very selective. Why is there no cross-architecture performance reported for other methods in Table 2?

---

> ### Author Response · Authors · 2025-11-29
> **Response to Reviewer kxaT [Part 1/4]**
>
> Thank you for your thoughtful and constructive feedback. We have uploaded a revised version of our paper incorporating all concrete revisions following your suggestions. We would like to further provide detailed point-by-point responses to address your concerns as follows:
>
> >W1 (1). The proposed method appears to be a marginal modification over SRe$^2$L.
>
> Thanks for the comment. We respectfully clarify that CV-DD is not a marginal extension of SRe$^2$L. Our method is designed to address two fundamental limitations of SRe$^2$L:
> 1. Limited Diversity, arising from the use of a single model during data synthesis.
> 2. Degraded Soft Labels, resulting from a distributional mismatch between the distilled and original datasets, introduced by regularization effects and optimization randomness, which undermines the quality and reliability of the soft labels generated by teacher models.
>
> To address these challenges, we introduce a Committee Voting mechanism that aggregates predictions and feature distributions from a diverse set of models during synthesis. This ensemble-based design substantially improves committee diversity and, in turn, promotes greater diversity in the distilled data. Furthermore, to improve the quality of the soft labels, we propose BSSL, a batch-specific soft labeling mechanism that leverages per-batch normalization to produce more stable and trustworthy supervisory signals, and thus improve the quality of the generated soft labels.
>
> Moreover, the effectiveness of these contributions is demonstrated by substantial performance gains (e.g., +6.4\% at IPC=10), which would not be achievable through marginal modifications. This confirms that CV-DD introduces meaningful methodological advances beyond SRe$^2$L.
>
> >W1 (2).  The core contribution, i.e., “Prior-based Voting”, is computationally expensive yet results in only marginal improvements, as shown in Table 4 (middle).
>
> Thanks for the comment. We kindly note that CV-DD is designed to be computationally efficient relative to existing ensemble-based methods, such as G-VBSM. CV-DD aligns only a single BN distribution during synthesis, whereas G-VBSM additionally aligns a convolutional-layer distribution, which results in G-VBSM needs  187.5 hours for distiling a IPC=50 dataset on ImageNet-1k, with CV-DD only requires  137.5 (including 84.7 hours for prior evaluation and 52.8 hours for distillation), which is 50 hours less.
>
> Furthermore, when compared to single-model distillation methods such as SRe$^2$L, we observe the following: SRe$^2$L requires 2.52 seconds per synthesized image and achieves 43.1% accuracy at IPC=10. Incorporating CV-DD into the SRe$^2$L framework increases the per-image optimization time to 3.82 seconds, i.e., 1.3 seconds slower, resulting in approximately 88.3 additional hours for distilling an IPC=10 dataset on a single RTX-4090 GPU (84.7 hours for prior evaluation and 3.6 hours for distillation). However, this additional computation yields a substantial performance improvement, achieving 49.5% accuracy, a +6.4% gain, a highly favorable trade-off given the magnitude of the accuracy increase.

---

> ### Author Response · Authors · 2025-11-29
> **Response to Reviewer kxaT [Part 2/4]**
>
> >W2. Several enhanced training tricks are adopted only for the proposed method and SRe$^2$L++, like "Smoothed Learning Rate & Smaller Batch Size", which is not applied to other competing methods, leading to potentially unfair comparisons.
>
> We thank the reviewer for the comment. We would like to clarify that the enhanced training tricks discussed in the paper are not exclusive to our proposed method. In fact, as summarized in the table below, all competing methods (EDC, CDA, RDED) also employ some of these tricks. Thus, these enhancements are not introduced only for CV-DD or SRe$^2$L++, but are used across baselines whenever applicable.
>
> It is worth noting that some methods do not adopt certain tricks not because they were intentionally excluded, but because they are incompatible with the method’s design. For example, RDED selects raw patches from the original dataset, which introduces high sample-level variance. This makes the optimization highly sensitive to batch statistics and therefore requires larger batch sizes for stability. Consequently, RDED cannot benefit from small-batch training, as smaller batches would amplify the variance and cause unstable updates. Thus, the absence of small-batch training in RDED reflects a methodological constraint, not a comparison bias.
>
> |     |  Small Batch Size | Smoothed Learning Rate |
> |:---|:---:|:----:|
> |   EDC  | ✓ | ✓|
> |   CDA  | ✓   | ×  |
> |   RDED  | ×  | ✓ |
>
> Among the baselines, CDA is the only method requiring additional consideration. To ensure fairness, we further evaluate CDA with the smoothed learning rate schedule, denoted as CDA++. As shown below, although CDA++ improves upon the original CDA, our proposed CV-DD consistently achieves the best performance across all IPC settings.
>
> |     |  CDA | CDA++  | CV-DD  |
> |:---|:---:|:----:|:----:|
> |   IPC=10  | 33.6 | 43.4  | **49.5**|
> |   IPC=50  | 53.5 | 54.3  | **59.5** |
>
>
> >W3. The comparison methods are limited. The main results only include two valid baselines, RDED and SR$^2$L, while another relevant method, CDA, which also follows a similar paradigm as SRe$^2$L, is missing under most experimental settings. In addition, several comparison methods are relatively outdated.
>
> Thanks for the comment. We would like to clarify a misunderstanding in the reviewer’s statement regarding the comparison baselines. In addition to RDED, CDA, and SRe$^2$L, our evaluation also includes GVBSM and EDC, where EDC represents one of the strongest uni-level dataset distillation methods. Therefore, it should not be considered outdated. With these baselines included, our evaluation setup is both comprehensive and competitive, directly benchmarking CV-DD against the strongest available approaches and clearly demonstrating the effectiveness of CV-DD.
>
> To further strengthen our comparison, we additionally include results against newly published methods in the following table, where we consistently observe improvements.
>
> |    | INFER [1] (ICLR 2025)  |  DELT [2] (CVPR 2025) | CV-DD  |
> |:---|:---:|:----:|:----:|
> |   IPC=10  | 36.3 |  46.1 | **49.5**|
> |   IPC=50  |  55.6|   59.2| **59.5** |
>
> >W4. The reported results are inappropriately cited. For instance, the MTT baseline is evaluated under an altered setting that deviates from its official configuration and appears to be intentionally adjusted in favor of the proposed method.
>
> We would like to clarify that the MTT results we report **are taken directly from the original paper (Table 1, CIFAR-100 and Tiny-ImageNet)** and are fully consistent with the published numbers. The reviewer’s concern appears to arise from a misunderstanding regarding the datasets and experimental configurations. Therefore, our citation and comparison are **both accurate and fair**.
>
> >W5. There are typos, e.g., in line 323, “Large scale dataset” should be written in lowercase.
>
> Thanks for pointing this out. We have corrected the typos in the revised version of the paper.

---

> ### Author Response · Authors · 2025-11-29
> **Response to Reviewer kxaT [Part 3/4]**
>
> >Q1. What is the architecture used in Fig. 1? Please specify it clearly.
>
> Thanks for the question. The architecture used in **Figure 1** is **ResNet-18**. We have updated the caption of Figure 1 accordingly in the revised version.
>
> >Q2. Regarding the Prior-based Voting in Section 3.4 and Algorithm 1, it appears that distillation and evaluation must be conducted separately for each model in the committee set, which will incur substantial computational overhead. In contrast, SRe2L also relies on pretrained models, but those are standard models that can often be assumed to be readily available.
>
> Thanks for raising this point. We would like to clarify that employing multiple models for distillation is inherently more time-consuming than single-model approaches, reflecting a natural time–performance trade-off. Although CV-DD includes pretraining, generation, and evaluation steps for prior performance estimation, its overall computational cost remains lower than that of other ensemble-based methods. This efficiency arises because CV-DD aligns only a single BN distribution, whereas methods such as G-VBSM and EDC additionally align a convolutional-layer distribution, resulting in significantly higher computational overhead.
>
> As shown in the table below (all experiments are conducted using a single RTX-4090), the total computation time for generating an IPC = 50 dataset is 137.5 hours (including 84.7 hours for prior evaluation and 52.8 hours for distillation), whereas G-VBSM requires 187.5 hours under the same setting. This demonstrates that, despite the additional prior evaluation stage, our approach achieves superior efficiency due to its significantly reduced per-iteration cost and fewer total optimization steps per image, while also delivering better performance than state-of-the-art ensemble methods.
>
> |     | Pretraining  | Generation | Evaluation | Peak GPU Usage | Total Time |
> |:---|:---:|:----:|:----:|:----:|:----:|
> |   ResNet-18  |   10.1 hours     |   1.3 hours  | 1.5 hours | 11.3 GB | 12.9 hours |
> |   ResNet-50  |   14.4  hours    | 4.4 hours |  1.5 hours   | 23.0 GB| 20.3 hours |
> |   ShuffleNetV2  |   9.4 hours     | 2.6 hours  | 1.5 hours | 7.6 GB | 13.4 hours|
> |   MobileNet-V2  |    10.2 hours    | 2.4 hours   | 1.5 hours| 11.4 GB| 14.1 hours|
> |   Densenet121  |   16.4 hours     |  6 hours   | 1.5 hours|19.5 GB |23.9 hours |
>
> Moreover, we compare the total generation time of G-VBSM across different IPC settings in the table below. As illustrated, this cumulative efficiency allows our framework to become increasingly advantageous as the target IPC grows. For instance, when distilling a 100-IPC dataset on ImageNet-1K, our method requires only 190.3 hours, whereas G-VBSM takes approximately 375 hours on a single RTX-4090. This analysis has been added to **Sec 4.4 (Table 5)** in the revised version.
>
> |    | G-VBSM | CV-DD  |
> |:---|:---:|:----:|
> |   IPC=50  |   187.5 hours    |  137.5 hours|
> |   IPC=100  |  375.0 hours |  190.3 hours|
> |   IPC=150  |  562.5 hours |  243.1 hours|
> |   IPC=200  |  750.0 hours |  295.9 hours|
>
>
>
> >Q3. Moreover, the MTT results in Table 2 are substantially lower than the original reported performance. What accounts for this degradation? Was it re-implemented using an ensemble strategy? If so, this deviates from the original setting and may distort the comparison, especially since the reported results are significantly lower than those reported in the original MTT paper.
>
> We emphasize that the reported values are exactly consistent with the original paper, and we strongly encourage the reviewer to refer to our response to Weakness 4 for detailed clarification.

---

> ### Author Response · Authors · 2025-11-29
> **Response to Reviewer kxaT [Part 4/4]**
>
> >Q4 (1). The ablation study is incomplete and fails to isolate the contribution of key components. Specifically, the impact of the proposed Prior-based Voting is not evaluated independently.
>
> Thanks. We first clarify that our ablation is complete and shows the contribution of key components, specifically, in table 4, we validate the effectiveness of prior-based voting, as summarized below, where we can see that incorporating prior voting consistently achieves better performance across different datasets.
>
> |     | SRe$^2$L++ (single Model) | CV-DD w/ Random Weighting  |  CV-DD w/ Equal Weighting| CV-DD w/ Prior Weighting |
> |:---|:---:|:----:|:----:|:----:|
> |   ImageNet-1k  | 43.1 |47.6| 48.2|  **49.5** |
> |   CIFAR-100  | 56.7  | 59.8| 60.7|  **61.8**|
>
> >Q4 (2). Moreover, several additional enhancements, such as Real Image Initialization, Data Augmentation, and Smoothed Learning Rate with Smaller Batch Size, are grouped together without showing their individual effects on performance.
>
> Thanks for this insightful question. To address this concern, we provide the performance gain for each component in the following table. We have added this experiment in **Appendix C.2** of the revision.
>
> | Data Augmentation |Real Data Initialization   | Small Batch Size | Smoothed Learning Rate | BSSL | Performance |
> |:---|:---:|:----:|:----:|:----:|:----:|
> |  ×  | ×    |    ×  |× |×| 23.3|
> |   ✓ |  ×   |   ×   | ×|×| 30.8|
> |   ✓ |  ✓   |   ×   | ×|×|  31.3 |
> |   ✓ |  ✓   |   ✓   | ×|×|  37.6   |
> |   ✓ |  ✓   |   ✓   | ✓|×|38.5|
> |   ✓ |  ✓   |   ✓   | ✓|✓|43.1|
>
>
> >Q4 (3). If these techniques are treated as default settings, they should be consistently applied to all compared methods (not only to SRe$^2$L) to ensure a fair and unbiased comparison.
>
> Thanks for the question. We kindly invite the reviewer to check our response to Weakness 2 for more details.
>
> >Q5. The reported results are very selective. Why is there no cross-architecture performance reported for other methods in Table 2?
>
> We would like to clarify that the reported results are **not selectively chosen**. *Instead, our intention is to ensure that stronger methods can be included for comparison considering the space constrains.* First, to address the reviewer's concern, we have included **G-VBSM** results in the **cross-architecture performance analysis** of our revised paper for completeness. As summarized in the table below, the performance of **G-VBSM** is even weaker than **SRe$^2$L++**, which is the primary reason we excluded it from the main comparison.
>
> | Model        | #Params | RDED | G-VBSM|EDC  | SRe$^2$L++ | **CV-DD** |
> |:-------------|:-------:|:----:|:----:|:-------:|:---------:|:----:
> | ShuffleNetV2 | 1.4M    | 19.6 | 16.0 |29.8 | 22.9    | **30.6**  |
> | MobileNetV2  | 3.4M    | 34.4 | 27.6 |45.0 | 37.1    | **45.6**  |
> | DenseNet121  | 8.0M    | 49.4 | 44.7 | –   | 46.7    | **54.7**  |
> | ResNet18     | 11.7M   | 42.0 | 31.4|48.6 | 43.1    | **49.5**  |
> | ResNet50     | 25.6M   | 49.7 | 35.4|54.1 | 47.3    | **57.0**  |
> | Swin-Tiny    | 28.0M   | 29.2 | - |38.3 | 28.3    | **39.2**  |
> | ResNet101    | 44.5M   | 48.3 | 38.2 |51.7 | 51.2    | **57.2**  |
> | RegNetX-8gf  | 39.6M   | 51.9 | 39.8 | –   | 53.4    | **60.9**  |
> | WRN-50-2     | 68.9M   | 50.0 | 48.6 | –   | 50.2    | **58.3**  |
>
> We also emphasize that **MTT** is difficult to scale up to **ImageNet-1K**, as also mentioned by many prior works, and thus we add TESLA [3] for furher comparison as shown in table below. To make it clearer, we originally did not include **G-VBSM** due to its **inferior performance** compared to **EDC** (to avoid redundancy from including weaker ones). Instead, we report the **best-performing methods** in each category, **RDED** for single-model distillation and **EDC** for multi-model ensemble distillation.
>
> |     | TESLA [3] | CV-DD   |
> |:---|:---:|:----:|
> |   IPC=10  |  17.8|  **49.5** |
> |   IPC=50  |  27.9|  **59.5**|
>
>
>
>
> References:
>
> [1] Xin Zhang, Jiawei Du, Ping Liu, and Joey Tianyi Zhou. "Breaking Class Barriers: Efficient Dataset Distillation via Inter-Class Feature Compensator." In ICLR 2025.
>
> [2] Zhiqiang Shen, Ammar Sherif, Zeyuan Yin, and Shitong Shao. "DELT: A Simple Diversity-driven EarlyLate Training for Dataset Distillation." In CVPR 2025.
>
>
> [3] Justin Cui, Ruochen Wang, Si Si, and Cho-Jui Hsieh. "Scaling Up Dataset Distillation to ImageNet-1K with Constant Memory." In ICML 2023.

---

### Official Review · Reviewer_RHkG · 2025-10-27

**Soundness:** 3
**Presentation:** 2
**Contribution:** 3
**Rating:** 6
**Confidence:** 3

**Summary:**

This paper presents Committee Voting for Dataset Distillation (CV-DD), a framework designed to generate diverse and representative distilled datasets by aggregating knowledge from multiple heterogeneous models. The method incorporates a Prior Performance Guided Voting Strategy to adaptively weight model contributions and a Batch-Specific Soft Labeling (BSSL) mechanism to mitigate distribution shifts between synthetic and real data. Experiments on CIFAR-10/100, Tiny-ImageNet, and ImageNet-1K indicate that CV-DD achieves competitive performance across various IPC settings and demonstrates strong cross-architecture generalization.

**Strengths:**

1. Committee voting is a novel approach for dataset distillation that improves data representativeness through prior performance–based weighting.
2. CV-DD achieves state-of-the-art results across datasets, e.g., 59.5% on ImageNet-1K (IPC=50) versus 56.5% for RDED, with strong cross-architecture generalization.
3. Ablations confirm the effectiveness of voting temperature, prior-guided voting, and BSSL in reducing distribution shift.

**Weaknesses:**

**Major:**

1. A large portion of the paper focuses on building a strong baseline SRe2L++, which already incorporates several optimizations from recent SOTA methods such as EDC and RDED. Since SRe2L++ itself performs at a very high level, the additional gain from the committee voting mechanism appears modest, reducing the overall sense of novelty.
2. Although the method reports better per-iteration efficiency than G-VBSM, the overall training pipeline is complex. It requires pretraining all committee models, running a time-consuming distillation–evaluation loop to assess prior performance, and maintaining multiple teacher models during training for gradient weighting. The high pretraining cost and runtime memory usage may limit its practicality on large-scale datasets such as ImageNet-1K.
3. The ablation study shows that increasing the number of experts N from 2 to 3 leads to performance degradation. This contradicts the intuition that more models should improve robustness, suggesting that the committee voting mechanism may have an upper limit or introduce redundancy, which restricts its scalability.

**Minor:**

1. The citation related to DD in Line 122 is incorrect.
2. The CIFAR-100 IPC-10/50 results in Table 2 are inconsistent with those in Table 1.
3. Figure 9 appears slightly blurred.

**Questions:**

1. Please include a clearer ablation study to isolate the contribution of the prior performance guided voting strategy over the SRe2L++ baseline. Specifically, compare (1) the original SRe2L++ (single model), (2) SRe2L++ with an equally weighted committee plus BSSL, and (3) the full CV-DD with prior-guided voting plus BSSL.
2. Beyond the per-iteration time reported in Table 3, provide a more detailed analysis of computational cost, including total pretraining time, prior performance evaluation time, and peak GPU memory usage, especially on ImageNet-1K. This would help readers assess the overall efficiency compared with single-model methods such as SRe2L++ or RDED.
3. Regarding the performance drop with larger N, please provide further analysis on model diversity and the voting mechanism. It would be useful to discuss whether two experts already provide sufficient diversity and whether adding a third causes gradient conflicts or unstable weighting.

---

> ### Author Response · Authors · 2025-11-29
> **Response to Reviewer RHkG [Part 1/3]**
>
> Thank you for your thoughtful and constructive feedback. We have uploaded a revised version of our paper incorporating all concrete revisions following your suggestions. We would like to further provide detailed point-by-point responses to address your concerns as follows:
>
> >W1. A large portion of the paper focuses on building a strong baseline SRe2L++, which already incorporates several optimizations from recent SOTA methods such as EDC and RDED. Since SRe2L++ itself performs at a very high level, the additional gain from the committee voting mechanism appears modest, reducing the overall sense of novelty.
>
> Thank you for the comment. We believe that constructing a strong baseline (SRe$^2$L++) with state-of-the-art performance is itself a meaningful contribution, as it provides a solid foundation for future research. In addition, the improvements achieved by our committee voting mechanism are non-trivial, for instance, we observe a notable +6.4% gain on ImageNet-1k (IPC=10) with ResNet-18, especially considering the difficulty of obtaining further improvements over an already strong baseline.
>
>
> >W2. Although the method reports better per-iteration efficiency than G-VBSM, the overall training pipeline is complex. It requires pretraining all committee models, running a time-consuming distillation–evaluation loop to assess prior performance, and maintaining multiple teacher models during training for gradient weighting. The high pretraining cost and runtime memory usage may limit its practicality on large-scale datasets such as ImageNet-1K.
>
> Thanks for raising this concern. To address it, we first report the total running time of G-VBSM for generating a 50-IPC dataset on a single RTX 4090. G-VBSM requires 4.32 ms per iteration and 4000 iterations to optimize a single image, resulting in approximately 17.3 seconds per image, or about **187.5 hours** in total to generate a 50-IPC dataset. Following the same estimation, our method requires only 52.8 hours for generating a 50-IPC dataset.
>
> We further provide a detailed breakdown of the total time required for Pretraining, Generation, and Evaluation of all committee members on ImageNet-1k, with all experiments conducted on a single RTX-4090. We observed that the original prior performance evaluation setting for ImageNet-1k was not optimal, as generating a 50-IPC dataset solely for evaluation is highly time-consuming. Instead, we found that using a 10-IPC distilled dataset for prior performance evaluation achieves nearly the same quality of distilled data, while substantially reducing the computational cost, as shown in the table below.
>
> |  |  Using **IPC=10** as Prior Performance Evaluation   | Using **IPC=50** as Prior Performance Evaluation |
> |:---|:---:|:---:|
> |IPC=10| 49.7| 49.5|
> |IPC=50| 59.3 | 59.5|
>
>
> As shown in the table below (all experiments are conducted using a single RTX-4090), the total computation time for generating an IPC = 50 dataset is 137.5 hours (including 84.7 hours for prior evaluation and 52.8 hours for distillation), whereas G-VBSM requires 187.5 hours under the same setting. This demonstrates that, despite the additional prior evaluation stage, our approach achieves superior efficiency due to its significantly reduced per-iteration cost and fewer total optimization steps per image, while also delivering better performance than state-of-the-art ensemble methods.
>
> |     | Pretraining  | Generation | Evaluation | Peak GPU Usage | Total Time |
> |:---|:---:|:----:|:----:|:----:|:----:|
> |   ResNet-18  |   10.1 hours     |   1.3 hours  | 1.5 hours | 11.3 GB | 12.9 hours |
> |   ResNet-50  |   14.4  hours    | 4.4 hours |  1.5 hours   | 23.0 GB| 20.3 hours |
> |   ShuffleNetV2  |   9.4 hours     | 2.6 hours  | 1.5 hours | 7.6 GB | 13.4 hours|
> |   MobileNet-V2  |    10.2 hours    | 2.4 hours   | 1.5 hours| 11.4 GB| 14.1 hours|
> |   Densenet121  |   16.4 hours     |  6 hours   | 1.5 hours|19.5 GB |23.9 hours |
>
> Moreover, we compare the total generation time of G-VBSM across different IPC settings in the table below. As illustrated, this cumulative efficiency allows our framework to become increasingly advantageous as the target IPC grows. For instance, when distilling a 100-IPC dataset on ImageNet-1K, our method requires only 190.3 hours, whereas G-VBSM takes approximately 375 hours on a single RTX-4090. This analysis has been added to **Sec 4.4 Table 5** in the revised version.
>
> |    | G-VBSM | CV-DD  |
> |:---|:---:|:----:|
> |   IPC=50  |   187.5 hours    |  137.5 hours|
> |   IPC=100  |  375.0 hours |  190.3 hours|
> |   IPC=150  |  562.5 hours |  243.1 hours|
> |   IPC=200  |  750.0 hours |  295.9 hours|

---

> ### Author Response · Authors · 2025-11-29
> **Response to Reviewer RHkG [Part 2/3]**
>
> >W3. The ablation study shows that increasing the number of experts N from 2 to 3 leads to performance degradation. This contradicts the intuition that more models should improve robustness, suggesting that the committee voting mechanism may have an upper limit or introduce redundancy, which restricts its scalability.
>
> We appreciate the reviewer for sharing this perspective. We would like to first clarify the distinct roles of the two components in the CV-DD framework.
>
> **1. Committee Size ($|S|$)**
> The committee size primarily controls the *diversity* of the distilled dataset. As shown in the left part of Table 5 (summarized below), we fix the number of experts to 2 and observe consistent performance improvements as $|S|$ increases. This confirms that a larger committee enhances dataset diversity, which in turn improves performance. Therefore, the committee size is mainly responsible for increasing the diversity of the distilled dataset.
>
> | R18 | R50 | D121 | SV2 | MBV2 | CIFAR-100 | ImageNet-1K |
> |:---:|:---:|:----:|:---:|:----:|:---------:|:-----------:|
> | ✓   |     |      |     |      |   56.7    |    43.1     |
> | ✓   | ✓   |      |     |      |   60.0    |    43.8     |
> | ✓   | ✓   | ✓    |     |      |   61.0    |    45.4     |
> | ✓   | ✓   | ✓    | ✓   |      |   61.5    |    48.3     |
> | ✓   | ✓   | ✓    | ✓   | ✓    |   61.8    |    49.5     |
>
> **2. Number of Experts**
> The number of experts controls how prior performance guides the voting strategy, emphasizing stronger recover model during optimization. However, increasing the number of experts beyond 2 can be counterproductive for two reasons:
>
> 1. **Dilution of strong recover model’s influence.**
>    With two experts, the best-performing recover model can receive a weighting ratio close to 0.9, while adding a third expert dilutes this ratio to around 0.5–0.6, weakening the impact of the strongest recover model.
>
> 2. **Gradient divergence.**
>    As shown below, we measure gradient consistency between the gradient derived from the best-performing recover model and that used in the image optimization step with two or three experts. The two-expert setup achieves higher gradient alignment, preserving knowledge from the best model while still benefiting from auxiliary recover models. In contrast, three experts lead to weaker consistency and reduced fidelity in the distilled dataset.
>
> | Experts | Gradient Consistency |
> |:--------:|:-------------------:|
> | 2 | 0.85 |
> | 3 | 0.42 |
>
> In short, the **committee size** enhances dataset diversity, while the **number of experts** leverages prior performance to guide optimization. Using two experts strikes an effective balance, retaining focus on the strongest model while incorporating complementary knowledge from others, thereby producing the most robust distilled dataset. We have added a more detailed analysis in **Sec 4.3 (Impact of the Number of Experts $N$)** to reflect this point.
>
>
>
> >W4. The citation related to DD in Line 122 is incorrect.
>
> Thanks for pointing this out. We have corrected the citation related to DD in Line 122 in the revised version.
>
> >W5. The CIFAR-100 IPC-10/50 results in Table 2 are inconsistent with those in Table 1.
>
> Thanks for this valuable comment. We would like to clarify that the results in Table 1 and Table 2 were obtained using different training length. Specifically, Table 1 uses 300 epochs for post-training for all datasets, whereas Table 2 uses 1000 epochs for post-training on CIFAR-100. This difference arises because the baseline methods compared in Table 1 (CDA, RDED) adopt different hyperparameter settings from those in Table 2 (EDC, G-VBSM). Therefore, it is expected that the CIFAR-100 results differ between the two tables.
>
> >W6. Figure 9 appears slightly blurred.
>
> We sincerely appreciate the reviewer’s comment. Figure 9 has been revised accordingly in the revised manuscript.

---

> ### Author Response · Authors · 2025-11-29
> **Response to Reviewer RHkG [Part 3/3]**
>
> >Q1. Please include a clearer ablation study to isolate the contribution of the prior performance guided voting strategy over the SRe2L++ baseline. Specifically, compare (1) the original SRe2L++ (single model), (2) SRe2L++ with an equally weighted committee plus BSSL, and (3) the full CV-DD with prior-guided voting plus BSSL.
>
> Thanks for the suggestion. We would like to clarify that SRe²L++ actually applies BSSL. The key difference between CV-DD and SRe$^2$L++ lies in that CV-DD leverages prior performance across a group of models, whereas SRe$^2$L++ utilizes only a single model. This comparison is already presented in the middle of Table 4 in the manuscript. As summarized below, CV-DD (w/ Equal weighting) corresponds to SRe$^2$L++ with an equally weighted committee plus BSSL. We observe that CV-DD (w/ Prior weighting) achieves the best overall performance, validating the effectiveness of prior-guided weighting. To further improve clarity, we have revised Table 3(b) to explicitly include the SRe$^2$L++ baseline in this comparison.
>
> |     | SRe$^2$L++ (single Model) | CV-DD w/ Random Weighting  |  CV-DD w/ Equal Weighting| CV-DD w/ Prior Weighting |
> |:---|:---:|:----:|:----:|:----:|
> |   ImageNet-1k  | 43.1 |47.6| 48.2|  **49.5** |
> |   CIFAR-100  | 56.7  | 59.8| 60.7|  **61.8**|
>
>
>
> >Q2. Beyond the per-iteration time reported in Table 3, provide a more detailed analysis of computational cost, including total pretraining time, prior performance evaluation time, and peak GPU memory usage, especially on ImageNet-1K. This would help readers assess the overall efficiency compared with single-model methods such as SRe2L++ or RDED.
>
> Thanks for the thoughtful question. Please refer to our response to Weakness 2 for a detailed discussion. We have also added more details in **Sec 4.4** to provide a comprehensive analysis of the computational cost.
>
>
> >Q3. Regarding the performance drop with larger N, please provide further analysis on model diversity and the voting mechanism. It would be useful to discuss whether two experts already provide sufficient diversity and whether adding a third causes gradient conflicts or unstable weighting.
>
> Thanks for this valuable comment and kindly invite the reviewer to refer to our response to Weakness 3 for further details.

---

### Official Review · Reviewer_tw2Q · 2025-10-30

**Soundness:** 2
**Presentation:** 2
**Contribution:** 2
**Rating:** 4
**Confidence:** 3

**Summary:**

This paper introduces Committee Voting for Dataset Distillation (CV-DD), a novel framework that enhances dataset distillation by combining predictions and feature distributions from multiple models (committee members), rather than depending on a single model. The authors also establish a stronger baseline, SRe2L++, and propose an improved criterion called Batch-Specific Soft Labeling (BSSL), which better aligns the distributions of real and synthetic images. The objective of CV-DD is to create smaller, high-quality synthetic datasets that preserve the essential characteristics of large real datasets while minimizing overfitting and reducing model-specific bias.

**Strengths:**

1. Ensemble-based methods in dataset distillation remain an emerging area of research.

2. The performance improvements reported in this paper are significant.

**Weaknesses:**

1. Although the proposed CV-DD framework enhances cross-model generalization by leveraging diverse architectures, its Batch-Specific Soft Labeling (BSSL) mechanism constrains the method to architectures that include Batch Normalization (BN) layers. This dependency limits the generalization and versatility of the approach when applied to models without BN components.

2. The visualization in Figure 6 suffers from poor distinguishability between curves due to the use of similar colors and marker styles for different data series. This makes it difficult for readers to clearly identify and interpret the individual trends. Additionally, the legend overlaps with the plotted data, further obscuring important details. It is recommended to improve the figure’s clarity by using more distinct color schemes or line styles and repositioning the legend to avoid overlapping with the data.

**Questions:**

1. **Comparison with SRe2L:**
 Could the authors include the performance results of SRe2L in the experiments? The reviewer is particularly interested in quantifying the improvement achieved by the enhanced SRe2L++ baseline over the original SRe2L.


2. **Ablation on Committee Composition:**
 The proposed method heavily depends on the quality and diversity of the committee members. However, the manuscript lacks a detailed discussion or analysis regarding this aspect. Could the authors include additional experiments that vary the composition or number of committee members to demonstrate the sensitivity and robustness of CV-DD to these factors?


3. **Dependence on Batch Normalization:**
 While CV-DD leverages the diversity of committee members, the proposed Batch-Specific Soft Labeling (BSSL) relies on architectures containing Batch Normalization (BN) layers. This dependence appears to limit the applicability of CV-DD to models without BN layers, such as Vision Transformers (ViTs) [1], which are widely adopted in modern computer vision. Could the authors discuss or experiment with how CV-DD might be extended to such architectures?


4. **Scalability with IPC:**
 How does the performance of CV-DD scale when the number of images per class (IPC) exceeds 50? Additional results or analysis in higher IPC regimes would help illustrate the scalability and practical limits of the proposed approach.


5. **Integration with Other Frameworks:**
 The proposed CV-DD presents an orthogonal contribution to existing literature and demonstrates strong potential for integration with other dataset distillation frameworks, such as RDED [2] and IGD [3]. Could the authors discuss or explore the feasibility and potential benefits of combining CV-DD with these frameworks?


[1] Alexey Dosovitskiy et al., An Image is Worth 16x16 Words: Transformers for Image Recognition at Scale, ICLR 2021

[2] Peng Sun et al., On the Diversity and Realism of Distilled Dataset: An Efficient Dataset Distillation Paradigm, CVPR 2024

[3] Mingyang Chen et al., Influence-Guided Diffusion for Dataset Distillation, ICLR 2025

---

> ### Author Response · Authors · 2025-11-29
> **Response to Reviewer tw2Q [Part 1/3]**
>
> Thank you for your thoughtful and constructive feedback. We have uploaded a revised version of our paper incorporating all concrete revisions following your suggestions. We would like to further provide detailed point-by-point responses to address your concerns as follows:
>
> >W1. Although the proposed CV-DD framework enhances cross-model generalization by leveraging diverse architectures, its Batch-Specific Soft Labeling (BSSL) mechanism constrains the method to architectures that include Batch Normalization (BN) layers. This dependency limits the generalization and versatility of the approach when applied to models without BN components.
>
> We appreciate the reviewer’s thoughtful comment. For architectures that do not employ BN layers (e.g., Vision Transformers (ViT)), we can easily adapt them to support BSSL. Specifically, we engineer a BN-ViT variant that replaces all LayerNorm layers with BatchNorm and inserts additional BN layers between the two linear layers in each feed-forward block, following the practice adopted in [1]. This simple modification allows non-BN architectures to benefit from BSSL without compromising their representational capacity. Therefore, the dependency on BN layers does not limit the general applicability of CV-DD, as it can be readily extended to a wide range of model architectures.
>
> To further validate this adaptation, we conduct an experiment using BN-ViT as the teacher model and compare the results with and without applying BSSL using CV-DD IPC=10. The BN-ViT+BSSL configuration achieves consistently higher performance, confirming that BSSL can be effectively extended to vit-based architectures. This simple modification allows non-BN architectures to benefit from BSSL without compromising their representational capacity. Therefore, the dependency on BN layers does not limit the general applicability of CV-DD, as it can be readily extended to a wide range of model architectures. We have included this experiment in **Sec 4.4 (Table 9).**
>
> |  Vit (w/o BSSL)|Vit (w BSSL)|
> |:----:|:----:|
> |   16.5   |  18.4    |
>
>
> >W2. The visualization in Figure 6 suffers from poor distinguishability between curves due to the use of similar colors and marker styles for different data series. This makes it difficult for readers to clearly identify and interpret the individual trends. Additionally, the legend overlaps with the plotted data, further obscuring important details. It is recommended to improve the figure’s clarity by using more distinct color schemes or line styles and repositioning the legend to avoid overlapping with the data.
>
> Thanks for pointing out the concern regarding the clarity of Figure 6. We have carefully revised the figure in the updated version by using more distinguishable color schemes, and we have repositioned the legend to avoid overlap with the data. Please check it out in our revision.
>
>
>
> >Q1. Comparison with SRe$^2$L: Could the authors include the performance results of SRe$^2$L in the experiments? The reviewer is particularly interested in quantifying the improvement achieved by the enhanced SRe$^2$L++ baseline over the original SRe$^2$L.
>
> Thanks for the suggestion. To address it, we present the quantitative improvements of SRe$^2$L++ over SRe$^2$L on both large- and small-scale datasets, as summarized below. We have added this comparison in **Appendix C.1 (Table 11)** of the revised version.
>
>
> **Comparison Between SRe$^2$L++ and SRe$^2$L on the Smaller Dataset (CIFAR-100)**
> |         |  SRe$^2$L |SRe$^2$L++|
> |:-------|:----:|:----:|
> |IPC10|   23.5    |  **56.7**    |
> |IPC50|    51.4   |  **66.6**    |
>
> **Comparison Between SRe$^2$L++ and SRe$^2$L on the Larger Dataset (ImageNet-1k)**
> |         |  SRe$^2$L |SRe$^2$L++|
> |:-------|:----:|:----:|
> |IPC10|   21.3    |  **43.1**    |
> |IPC50|   46.8    |  **57.6**    |

---

> ### Author Response · Authors · 2025-11-29
> **Response to Reviewer tw2Q [Part 2/3]**
>
> >Q2. Ablation on Committee Composition: The proposed method heavily depends on the quality and diversity of the committee members. However, the manuscript lacks a detailed discussion or analysis regarding this aspect. Could the authors include additional experiments that vary the composition or number of committee members to demonstrate the sensitivity and robustness of CV-DD to these factors?
>
> Thanks for raising this important point. We clarify that this experiment has already been included in the left part of Table 5 in the manuscript. As summarized in the table below, we observed that CV-DD is not sensitive to the performance of individual models and remains highly robust to these models. In our prior performance analysis, ResNet-50 (R50) exhibited the lowest standalone prior performance; however, adding R50 to ResNet-18 (R18), which achieved the highest prior performance, still resulted in consistent gains of +3.3 on CIFAR-100 and +0.7 on ImageNet-1K. Furthermore, we observed a positive correlation between the distillation performance and the number of models (i.e., $|S|$) in the committee. This improvement can be attributed to the fact that a larger $|S|$ introduces greater diversity into the distilled dataset, as formalized in Theorem 1, which in turn leads to better distillation performance.
>
>
> | R18 | R50 | D121 | SV2 | MBV2 | CIFAR-100 | ImageNet-1K |
> |:---:|:---:|:----:|:---:|:----:|:---------:|:-----------:|
> | ✓   |     |      |     |      |   56.7    |    43.1     |
> | ✓   |  ✓  |      |     |      |   60.0    |    43.8     |
> | ✓   |  ✓  |  ✓   |     |      |   61.0    |    45.4     |
> | ✓   |  ✓  |  ✓   |  ✓  |      |   61.5    |    48.3     |
> | ✓   |  ✓  |  ✓   |  ✓  |  ✓   |   61.8    |    49.5     |
>
>
>
> >Q3. Dependence on Batch Normalization: While CV-DD leverages the diversity of committee members, the proposed Batch-Specific Soft Labeling (BSSL) relies on architectures containing Batch Normalization (BN) layers. This dependence appears to limit the applicability of CV-DD to models without BN layers, such as Vision Transformers (ViTs), which are widely adopted in modern computer vision. Could the authors discuss or experiment with how CV-DD might be extended to such architectures?
>
> Thanks for this insightful point. We clarify that BN layers can be added to models that do not originally include them. Please refer to our response to Weakness 1 for further details.
>
> >Q4. Scalability with IPC: How does the performance of CV-DD scale when the number of images per class (IPC) exceeds 50? Additional results or analysis in higher IPC regimes would help illustrate the scalability and practical limits of the proposed approach.
>
> Thanks for raising this insightful question. We include below a comparison of CV-DD with SRe$^{2}$L++ and RDED under higher IPC settings. As shown in the table, CV-DD consistently outperforms both baselines across all configurations, demonstrating its strong scalability and effectiveness in higher IPC regimes.
>
> |     | RDED |CV-DD |
> |:---:|:---:|:----:|
> |   IPC=60  |  58.0   |  **60.5**  |
> | IPC=70   |  59.0    | **61.7**    |

---

> ### Author Response · Authors · 2025-11-29
> **Response to Reviewer tw2Q [Part 3/3]**
>
> >Q5. Integration with Other Frameworks: The proposed CV-DD presents an orthogonal contribution to existing literature and demonstrates strong potential for integration with other dataset distillation frameworks, such as RDED and IGD. Could the authors discuss or explore the feasibility and potential benefits of combining CV-DD with these frameworks?
>
> Thanks for raising this thoughtful question. Following your suggestion, we incorporate the idea of **CV-DD** into **RDED**, where the original RDED employs only a single model to select patches from the dataset. Its selection function is defined as follows:
>
> $$
> \xi \_{i,⋆} = \arg\max \_{\xi \_{i,k} \sim p(\xi_ {i,k} | \hat{x}_i)}
> \left[-\ell\big(\phi \_{\theta \_T}(\xi \_{i,k}), \phi \_h(\xi \_{i,k})\big)\right].
> $$
>
> Specifically, we extend this formulation by using multiple models to select patches from the original dataset and weighting their losses according to each model’s prior performance, as shown below:
>
> $$
> \xi \_{i,⋆} = \arg\max \_{\xi \_{i,k} \sim p(\xi \_{i,k} | \hat{x} \_i)}
> \left[-\sum \_{m=1}^{n} w \_m  \ell\big(\phi \_{\theta_m}(\xi \_{i,k}), \phi \_h(\xi \_{i,k})\big)
> \right],
> $$
>
> where $w_m = \frac{\exp(\alpha_m / T)}{\sum_{j=1}^{n} \exp(\alpha_j / T)}$. We report the performance of the vanilla RDED and its CV-DD–enhanced variant in the table below. We observe consistent performance improvements when integrating CV-DD with existing frameworks such as RDED, demonstrating its orthogonal and complementary nature. These results highlight that CV-DD can serve as a plug-in component to enhance other dataset distillation paradigms by leveraging model diversity and prior-guided voting. This confirms the feasibility and clear benefits of combining CV-DD with various frameworks, further broadening its applicability in the dataset distillation literature. We have included this analysis in **Sec 4.4 Table 6.**
>
> |     | RDED  |RDED w/ Equal Voting | Rded w/ Prior Voting (**CV-DD**) |
> |:---|:---:|:----:|:----:|
> |   IPC=10  |   42.0 |   43.2  | **44.8**|
> |   IPC=20  |    47.9    |   48.5  | **49.7** |
> |   IPC=30  |   51.7    | 52.1      |   **53.2**  |
>
>
> ----
> References:
>
> [1] Zhuliang Yao, Yue Cao, Yutong Lin, Ze Liu, Zheng Zhang, and Han Hu. "Leveraging batch normalization for vision transformers." In ICCV 2021.

---

### Author Response · Authors · 2025-12-03
**Summary for Reviews and Rebuttals**

Dear Reviewers and AC,

We would like to express our sincere gratitude for the time, effort, and thoughtful comments that all reviewers dedicated to our submission. We greatly appreciate the constructive feedback and are thankful that various strengths of our work were recognized, including the effectiveness of our framework [**Reviewers tw2Q, RHkG, 4Bi4**], the novelty of applying committee voting to Dataset Distillation [**Reviewer RHkG**], the clarity of our motivation and insights [**Reviewers 4Bi4, kxaT**], the soundness of our technical design [**Reviewer 4Bi4**], and the theoretical foundations supporting our approach [**Reviewer 4Bi4**].

**Reviewers** **4Bi4** and **RHkG** provided positive assessments, and we have provided comprehensive rebuttals and additional experiments addressing all their raised concerns.

**Reviewer tw2Q** initially gave a score of 4, raising 6 concerns: (1) BSSL for non-BN model; (2) unclear Figure 6; (3) comparison with SRe2L; (4) ablation on committee composition; (5) scalability with IPC; and (6) integration with other frameworks.

1. To address Concern 1, we have constructed BN-ViT variants and demonstrated that BSSL can generalize to models without BN once BN layers are manually inserted.
2. To address Concern 2, we have refined the legends and improved the clarity of Figure 6 in the revised manuscript.
3. To address Concern 3, we have added a comparison between SRe$^2$L and SRe$^2$L++.
4. To address Concern 4, we have clarified that the corresponding results are already included in the original manuscript.
5. To address Concern 5, we have conducted additional experiments under larger IPC settings.
6. To address Concern 6, we have reported further results on integrating committee voting into RDED, demonstrating its effectiveness beyond our framework.

**Reviewer kxaT** initially gave a score of 2, raising 6 concerns: (1) marginal modification over SRe$^2$L; (2) potentially unfair comparisons; (3) limited comparison methods; (4) the reported results are inappropriately cited; (5) computational overhead; and (6) selective results.

1. For Concern 1, we have clarified our framework's main contributions beyond SRe$^2$L.
2. For Concern 2, we have explained that the evaluation is fair since competing methods use the same techniques.
3. For Concern 3, we have added more recent SOTA baselines.
4. For Concern 4, we have confirmed that the results match the original paper.
5. For Concern 5, we have reported total clock time and compared it with ensemble-based methods to show our efficiency.
6. For Concern 6, we have explained why G-VBSM was excluded due to poor performance and added additional TESLA for further comparison.

We sincerely appreciate the reviewers' thoughtful suggestions and feedback during the review process, and we hope that our summary clarifications above and detailed responses below have addressed all reviewers' concerns. We have also uploaded a revision of our manuscript that incorporates all reviewer comments.

Best,

Authors of submission 13004

---

### Meta-Review · Area_Chair_6XvY · 2025-12-22

**Summary:**

The major concerns are about the novelty of the work and the lack of ablation study to separate the contribution of each component in the framework. Some reviewers worried that the novel part (prior-based voting) does not contribute much to the performance improvement. Other concerns include sensitivity of committee design, the fairness of comparisons, the lack of baselines, computational/scalability cost of the pipeline, and the breadth of evaluation, including cross-architecture experiments. Some of these concerns were addressed by the rebuttal.

**Reviewer Concerns:**

The major concern about the novelty is not fully addressed. The author highlighted the performance gain of prior-based voting at IPC=10 with ResNet 18, but this evidence alone is not sufficient to convincingly demonstrate the overall contribution of the proposed voting mechanism. It will be sufficient if the performance improvement is shown under different IPCs, architectures, and datasets, like in the main results table. For other concerns, the authors made a strong effort to address the reviewers’ questions by including additional experiments. However, due to the time limit, these new results are limited in scale or scope. Therefore, several concerns remain only partially addressed. A more thorough experimental study, which takes longer time, would be necessary to fully resolve the issues.

**Reviewer Scores:**

Reviewer tw2Q may increase the score. Reviewer 4Bi4 and RHkG are likely to keep the positive scores. Reviewer kxaT may not increase the score as the major concern about novelty does not appear to be fully addressed.

---

### Decision · Program_Chairs · 2026-01-26

Reject